# Temporospatial hierarchy and allele-specific expression of zygotic genome activation revealed by distant interspecific urochordate hybrids

Jiankai Wei[1,2,3,8], Wei Zhang[1,8], An Jiang[1,8], Hongzhe Peng[1,8], Quanyong Zhang[4,8], Yuting Li[1], Jianqing Bi[1], Linting Wang[5], Penghui Liu[1], Jing Wang[1], Yonghang Ge[1], Liya Zhang[4], Haiyan Yu[1], Lei Li [5], Shi Wang [1,2], Liang Leng [6] ✉, Kai Chen [4,7] ✉ & Bo Dong [1,2,3] ✉

Zygotic genome activation (ZGA) is a universal process in early embryogenesis of metazoan, when the quiescent zygotic nucleus initiates global transcription. However, the mechanisms related to massive genome activation and allele-specific expression (ASE) remain not well understood. Here, we develop hybrids from two deeply diverged (120 Mya) ascidian species to symmetrically document the dynamics of ZGA. We identify two coordinated ZGA waves represent early developmental and housekeeping gene reactivation, respectively. Single-cell RNA sequencing reveals that the major expression wave exhibits spatial heterogeneity and significantly correlates with cell fate. Moreover, allele-specific expression occurs in a species- rather than parent-related manner, demonstrating the divergence of cis-regulatory elements between the two species. These findings provide insights into ZGA in chordates.

Although metazoans exhibit divergent cleavage patterns, the underlying molecular process that initializes developmental programming is highly conserved with few exceptions, even in plants. The zygotic genome is initially quiescent following fertilization, with several developmental genes being transcribed during cleavage. Later on, widespread zygotic genome activation (ZGA) occurs coincident with housekeeping gene reactivation. Stepwise ZGA and transient silencing of housekeeping genes have been documented in several model species, suggesting that this process has been preserved under extreme selection pressure and is important in embryogenesis[1]. Knowledge of this process has largely been acquired from studies of *Drosophila*, *Danio rerio*, and *Xenopus*, which show mid-blastula transition during the major ZGA that co-occurs with the transition from synchronous division to the prolonged asynchronous division[2].

Several models have been proposed to explain the mechanisms of ZGA; these include the nuclear-to-cytoplasmic (N/C) ratio model[3,4],

[1]Fang Zongxi Center for Marine EvoDevo, MoE Key Laboratory of Marine Genetics and Breeding, College of Marine Life Sciences, Ocean University of China, Qingdao 266003, China. [2]Laboratory for Marine Biology and Biotechnology, Qingdao Marine Science and Technology Center, Qingdao 266237, China. [3]MoE Key Laboratory of Evolution and Marine Biodiversity, Institute of Evolution and Marine Biodiversity, Ocean University of China, Qingdao 266003, China. [4]State Key Laboratory of Primate Biomedical Research and Institute of Primate Translational Medicine, Kunming University of Science and Technology, Kunming, Yunnan 650500, China. [5]National Center of Mathematics and Interdisciplinary Sciences, Academy of Mathematics and Systems Science, Chinese Academy of Sciences, Beijing 100190, China. [6]Institute of Herbgenomics, Chengdu University of Traditional Chinese Medicine, Chengdu 611137, China. [7]Present address: Southern Marine Science and Engineering Guangdong Laboratory (Guangzhou), No. 1119 Haibin Rd, Nansha Dist., Guangzhou 511458, China. [8]These authors contributed equally: Jiankai Wei, Wei Zhang, An Jiang, Hongzhe Peng, Quanyong Zhang. ✉e-mail: lleng@icmm.ac.cn; chen_kai@gmlab.ac.cn; bodong@ouc.edu.cn

activator-accumulation model[5], and chromatin state dynamics model[6,7]. The N/C ratio model proposes that an increase in the N/C ratio could titrate the repressor to relieve transcriptional repression, whereas the activator accumulation model considers that essential factors must be accumulated to enable transcription, and the chromatin state model emphasizes the ability of the transcriptional machinery to access the DNA. In addition to these models, the maternal-clock model presumes that fertilization initiates a biochemical cascade that serves as a molecular timer[8]. Comprehensive maturity of nuclear pore complexes are also proposed to regulate ZGA[9]. These models are not mutually exclusive, and it is becoming increasingly clear that multiple processes regulate ZGA timing in a coordinated manner[10,11].

ZGA processes are distinct according to different developmental paradigms[11], and the regulation of ZGA in non-model animals remains poorly understood. Thus, the presumption of the initial state of ZGA during evolution will help in understanding the intrinsic interaction of different ZGA models. Therefore, research on ZGA in a range of organisms, especially those associated with pivotal transition points during animal evolution, would provide additional insights into the evolution and adaptation of the molecular strategies underlying ZGA.

Allelic expression imbalance or allele-specific expression (ASE) often occurs during ZGA. Studies have shown that two sets of chromosomes are not functionally equivalent during embryonic development[12,13]. ASE is affected not only at various levels from chromatin state to transcription and splicing, but also by the rate of protein translation[14]. To date, high-throughput sequencing strategies using reciprocal hybrids to identify ASE genes are mainly based on heterozygous single-nucleotide polymorphisms (SNPs)[15]; however, this approach is limited by known SNPs sites. Hence, methods that can accurately infer the parental origin effects at the molecular level are still required.

Ascidians are one of the closest living relatives of vertebrates and have been used in embryonic research for more than a century, with extensive study of the initiation of the developmental programming in ascidian embryos[16–19]. The cell lineages of multiple ascidian species have been well characterized and their genomes sequenced[20–23]. Two co-occurring solitary species, *Ciona robusta* and *C. savignyi*, have been widely used in gene regulatory network studies to understand chordate embryogenesis and evolution. Despite their conserved embryogenesis patterning, these species diverged approximately 122 (± 33) Mya[24], which is similar to the divergence time of humans and kangaroos[25].

Herein, we used *C. robusta* and *C. savignyi* hybrids as models to study the temporal dynamics of ZGA. We also assessed the spatial reactivation of housekeeping genes during ZGA at a single-cell resolution. Furthermore, we revealed and experimentally verified the gene expression preference between maternal and paternal transcripts during ZGA. Our results improve our understanding of the dynamics of ZGA and provide insights into the regulatory mechanisms of this critical process in urochordates.

## Results

### Embryogenesis of hybrids from two deeply diverged *Ciona* species
Both *C. robusta* and *C. savignyi* are tunicate species with similar morphology that are broadly distributed across the North Pacific Ocean (Fig. 1a) but diverged ~122 Mya (Fig. 1b), which is larger than the distance of any two extant placentals[24]. The interfertility of these two highly divergent species was reported in a previous study[26]. Although the two species are morphologically similar, their genomic sequences are highly divergent. The sequence discrepancy of the orthologous genes between these two species is ~14.3% ± 3.9% based on sequence alignment (Supplementary Fig. 1), enabling alleles to be assigned to either male or female parents in hybrid experiments.

We first used the dechorionized eggs of the two species to assess the developmental capabilities of the hybrid embryos[27]. The embryogenesis of both successful crosses involved a similar developmental time and morphology until the tailbud stage (Fig. 1c). Although one direction of the hybrid embryos (Cs♀ × Cr♂, defined as reverse cross) had a higher development failure rate than that of the cross (Cr♀ × Cs♂, defined as forward cross) at the early neurula stage (Supplementary Data 1, Supplementary Fig. 2), there was no visible difference in the embryogenesis process between forward and reverse cross embryos before the tailbud stage, which is distant from the ZGA sampling window (from 1-cell stage to gastrula stage). The embryo development observations are provided (Supplementary Fig. 2). Bulk RNA sequencing (RNA-seq) and single-cell RNA-seq were then performed to examine their expression profiles (Fig. 1d).

### Temporal dynamics of ZGA from hybrid ascidian embryos
Illumina short reads of bulk RNA-seq from each stage were mapped to the genomes of *C. robusta* and *C. savignyi*. The reads mapped to paternal genomes evenly increased from the 8-cell stage, based on the heat map (Fig. 2a). Then the ratio of aligned reads to gene regions was calculated. For both forward and reverse cross embryos, >99.99% of the reads were unambiguously mapped to the maternal genes in samples of 1-, 2-, and 4-cell stages, whereas paternal transcripts were detected from the 8-cell stage, and their proportion increased from the 16-cell to 32-cell stage (from 0.02% to 0.16% and from 0.02% to 0.22% in forward and reverse cross embryos, respectively). Additionally, the paternal reads ratio increased rapidly from the 64-cell to 112-cell stage (from 0.53% to 2.29% and from 1.05% to 5.33% in forward and reverse cross embryos, respectively) (Fig. 2b, Supplementary Data 2). The disproportionate transcription of maternal and paternal genes was diminished at the early-neurula and late-neurula stages. For example, at the late-neurula stage, 13.65% and 26.90% reads were mapped to paternal genes in forward and reverse cross embryos, respectively. However, the proportion of maternal reads remained higher than that of the paternal reads.

The gene expression levels were then measured using fragments per kilobase of genes per million mapped reads (FPKM) values, wherein a minimum threshold of 0.1 was applied to filter expressed genes. The first expression of paternal genes occurred at the 8-cell stage in both forward and reverse cross embryos. These genes were *foxA2*, *regulator of G-protein signaling 8* (*rgs8*), and *pbx1* in forward cross embryos, and *foxA2*, *Sox2*, *SH3 domain-binding protein 4* (*SH3BP4*), *DNA-directed RNA polymerase I subunit RPA1* (*RPA1*), *brachyury*, and *rgs8* in reverse cross embryos. The number of paternal genes that were transcribed for the first time showed an increase, especially from the 16- to 32-cell stage and from the 64- to the 112-cell stage (Fig. 2c), consistent with the previous study[28]. In addition, the intronic reads ratio showed a similar trend with the proportion of paternal genes (Supplementary Fig. 3, Supplementary Data 2). Principal component (PC) analysis divided the bulk RNA-seq samples into four major clusters. PC1 divided them into forward and reverse cross samples, and PC2 divided them at 112-cell stage (Supplementary Fig. 4). The distinction between the 64- and 112-cell stages was consistent with the results of the ratio of paternal genes and the de novo transcribed paternal gene number.

To confirm the gene expression characteristics and whether maternal orthologs were also transcribed as paternal orthologs, we further performed cleavage under targets and tagmentation (CUT&Tag) experiments for RNA polymerase II (Pol II) and histone H3 lysine 27 acetylation (H3K27ac, a histone modification associated with active enhancer) in both forward and reverse cross embryos at 16-, 32-, 64-, and 112-cell and early neurula stages. In the reverse cross embryos, H3K27ac signals were significantly increased from 64- to 112-cell stage in both paternal and maternal genomes (Fig. 2d). Pol II signals were detected at both parental orthologs for the early zygotic genes and the

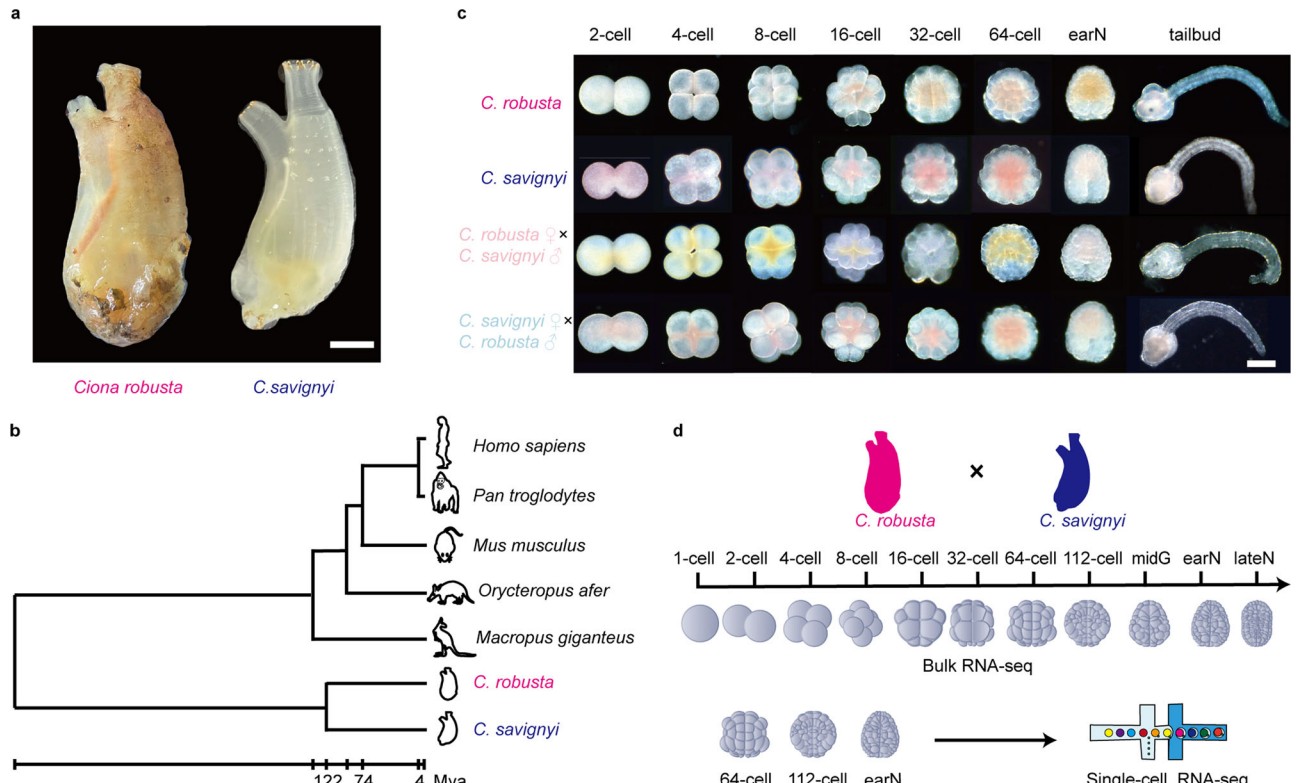

**Fig. 1 | Generating hybrid and gene expression profiles from two divergent *Ciona* species. a** Morphology of adult *C. robusta* (left) and *C. savignyi* (right). Scale bar: 1 cm. **b** Phylogenetic position and divergence time estimation of *Ciona*. Divergence times are displayed below the phylogenetic tree. **c** Different developmental stages of four cross ascidians, including self-crossing *C. robusta*, self-crossing *C. savignyi*, *C. savignyi* as a male parent cross with *C. robusta* (forward cross), and *C. robusta* as a male parent cross with *C. savignyi* (reverse cross). Eight developmental stages were observed including 2-, 4-, 8-, 16-, 32-, and 64-cell, early-neurula (earN), and tailbud stages. Scale bar: 50 μm. The cross experiment was independently performed at least three times. **d** Overview of the gene expression profiling strategy. Forward and reverse cross embryos collected for bulk RNA sequencing included 1-, 2-, 4-, 8-, 16-, 32-, 64-, and 112-cell, mid-gastrula (midG), earN, and late-neurula (lateN) stages. Cross embryos at the 64- and 112-cell and earN stages were collected for single-cell RNA sequencing.

reactivated housekeeping genes (Supplementary Fig. 5), which was consistent with RNA-seq results, and supported the assumption of using paternal transcripts for studying housekeeping gene reactivation with minimal artefacts derived from the interspecific cross. Gene expression and regulatory profiles revealed that the initiation of ZGA in *Ciona* began at the 8-cell stage, with the minor wave mainly occurring from the 16- to 32-cell stage and the major wave from the 64- to 112-cell stage during ascidian embryogenesis.

Weighted correlation network analysis (WGCNA) showed that genes were clustered into five modules in either forward or reverse cross embryos (Supplementary Fig. 6). A heatmap of paternal gene expression showed that the paternal genes in module 1 were mostly upregulated from the 16- to 32-cell stage, whereas the paternal genes in modules 2 and 3 were mostly upregulated from the 64- to 112-cell stage in both directional crosses (Fig. 2e, f). Gene ontology (GO) enrichment analysis of each module showed that the paternal genes in module 1 (minor wave) were related to GO terms of biological processes including regulation of transcription, regulation of nucleic acid, and regulation of RNA biosynthetic process, implying their regulatory effects on the major wave genes, whereas the major wave were corresponded with the housekeeping gene reactivation (Supplementary Fig. 6). The orthologous analysis between parental genes and overlap analysis of both directional crosses showed conservation of genes in module 1 and module 2–3 (Supplementary Data 3).

Overall, the gene expression profiles and genome-wide H3K27ac and Pol II occupancy analysis of hybrid embryos indicate two waves of ZGA, especially from the 16- to 32-cell stage and from the 64- to the 112-cell stage. The expression of genes in the two waves demonstrated to be potentially coordinated regulation relationships.

## Asynchronized spatial pattern of ZGA revealed by single-cell transcriptome

The development with invariant lineage in *Ciona* embryos provides a unique opportunity to study ZGA and test prevailing models simultaneously at single-cell resolution. Asynchronized division of *Ciona* initiates at the 16-cell stage, which is earlier than the major ZGA (coincident with housekeeping gene reactivation)[29]. Therefore, we used single-cell RNA-seq to examine the reactivation of paternal housekeeping genes of 64-cell, 112-cell, and early neurula stage embryos (i.e., when the major ZGA wave occurred). Housekeeping genes examined in this study were maternally expressed and also have been reactivated at the late-neurula stage, which were also followed the criteria as essential genes required for cellular existence and stably expressed irrespective of developmental stage.

In total, 8,357 single-cell transcriptomes were obtained after filtering from forward cross hybrids. Cells from 64-cell, 112-cell, and early-neurula stage embryos of forward cross hybrids were clustered into 9, 12, and 19 clusters, respectively, using uniform manifold approximation and projection (UMAP) for dimension reduction method. Cell cluster identities were annotated as endoderm, notochord, mesenchyme, muscle, nerve cord, germ, neural, and epidermis (Fig. 3a, d; Supplementary Figs. 7–9; Supplementary Data 4–6) by aligning with the published gene expression patterns[30,31].

To compare the ZGA in each cell cluster, we calculated the transcript ratio of the paternal housekeeping genes to all genes in each cell

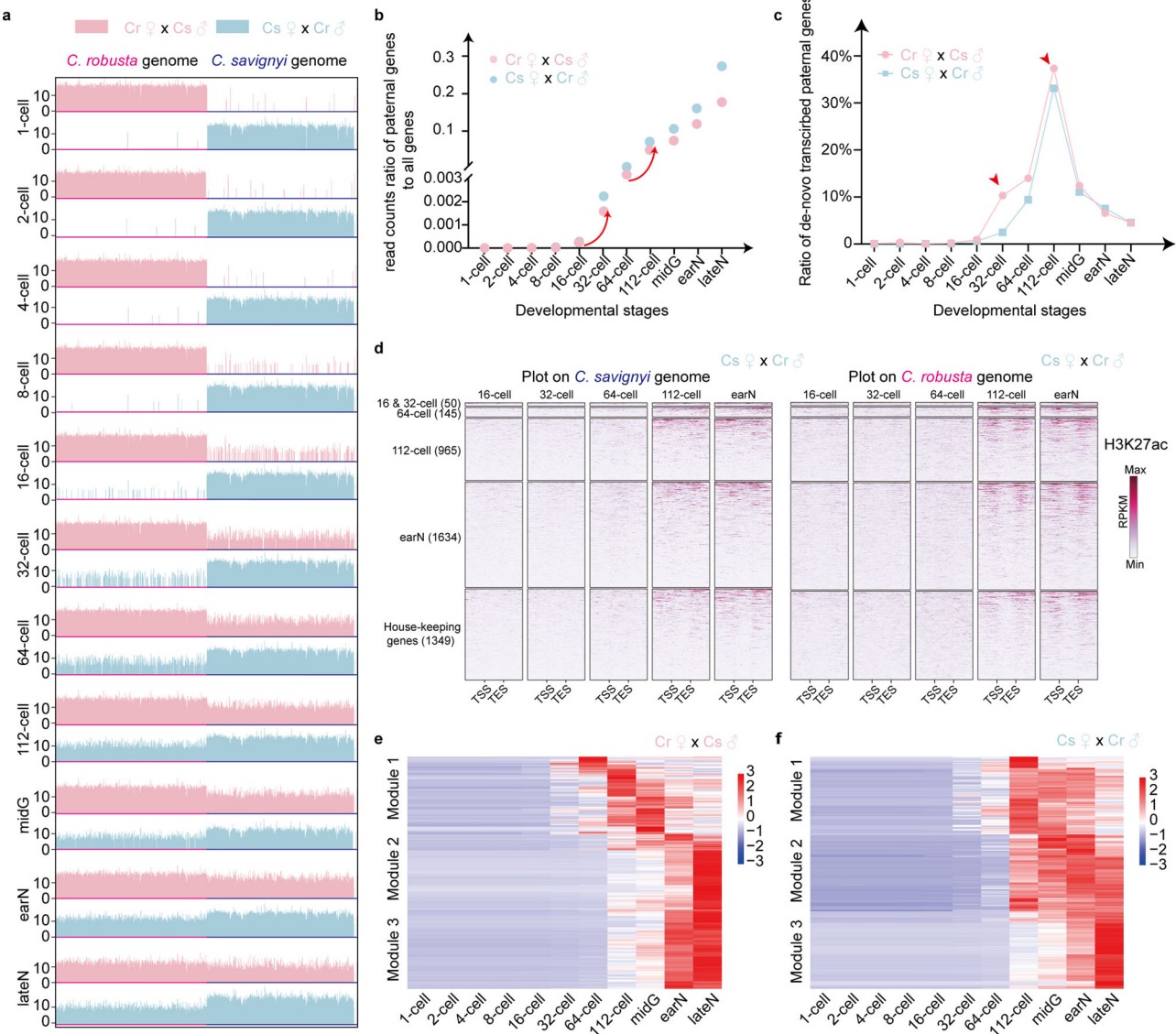

**Fig. 2 | Two coordinated waves of ZGA during ascidian embryogenesis.**
**a** Illustration of reads mapped to *C. robusta* and *C. savignyi* genomes at different developmental stages. Horizontal and vertical axes represent genome position and $\log_2$-transformed read number in each window, respectively. The bright red and bright blue indicated *C. robusta* and *C. savignyi*, while the light red and light blue indicated the forward and reverse cross samples, respectively. **b** read count ratios of paternal genes to all genes at different developmental stages. The red arrows indicated the two waves during ZGA. **c** Ratio of number of de-novo transcribed paternal genes to number of all paternal genes at different developmental stages. The red arrows indicated the minor and major ZGA which corresponded with the initiation of zygotic gene expression and housekeeping gene reactivation,

respectively. **d** Heat map of H3K27ac CUT&Tag signals across activated genes and housekeeping genes in reverse cross hybrid embryos. Each line shows the normalized signals for a gene from -2k bp above TSS to +2k bp below TES. The number in parentheses indicated the de-novo transcribed gene number in each stage.
**e** Heatmap of paternal gene expression in different modules in forward cross embryos. Module 1 included minor wave genes, the expression of which increased from the 16- to 32-cell stage, whereas modules 2 and 3 included major wave genes, the expression of which increased from the 64- to 112-cell stage. **f** Heatmap of paternal gene expression in different modules in reverse cross embryos. The description of the modules given in (**e**) also applies here. Source data are provided as a Source Data file.

type. At the 64-cell stage, the endoderm cells had the highest paternal transcript ratio, followed by notochord cells (Fig. 3b, c; Supplementary Fig. 10a, b; Supplementary Data 7). At the 112-cell stage, endoderm cells and notochord cells also had the top two highest paternal transcript ratios (Fig. 3e, f; Supplementary Fig. 10c, d; Supplementary Data 7). At the early-neurula stage, the paternal transcripts increased in nerve cord and mesenchyme B lineage cells, and the differences in the paternal transcript ratios of each lineage were decreased (Supplementary Fig. 11; Supplementary Data 7). The single-cell RNA-seq for reverse cross at 112-cell stage was also analyzed (Supplementary Fig. 12; Supplementary Data 8) and indicated similar results (Supplementary Figs. 13, 14; Supplementary Data 9).

To further validate the asynchronized ZGA, we performed whole-mount in situ hybridizations of paternal housekeeping genes in forward cross embryos and self-crossed *C. savignyi* embryos at 32-, 64- and 112-cell stages. Enriched expression signals were detected for 60 S ribosomal protein L11 (*6OSL11*), 60 S ribosomal protein L13 (*RPL13*), transport protein SEC61 (*SEC*), KRR1 small subunit processssome component (*KRR*), and heterogeneous nuclear ribonucleoprotein D0 (*HNRN*) in endoderm cells from the 32- to 112-cell stage in forward cross embryos, while nonspecific enriched signals were detected in self-crossed *C. savignyi* embryos (Fig. 3g). No detectable signals using sense probes (Supplementary Fig. 15), indicated the specificity of the anti-sense probes. Overall, these results indicated that zygotic

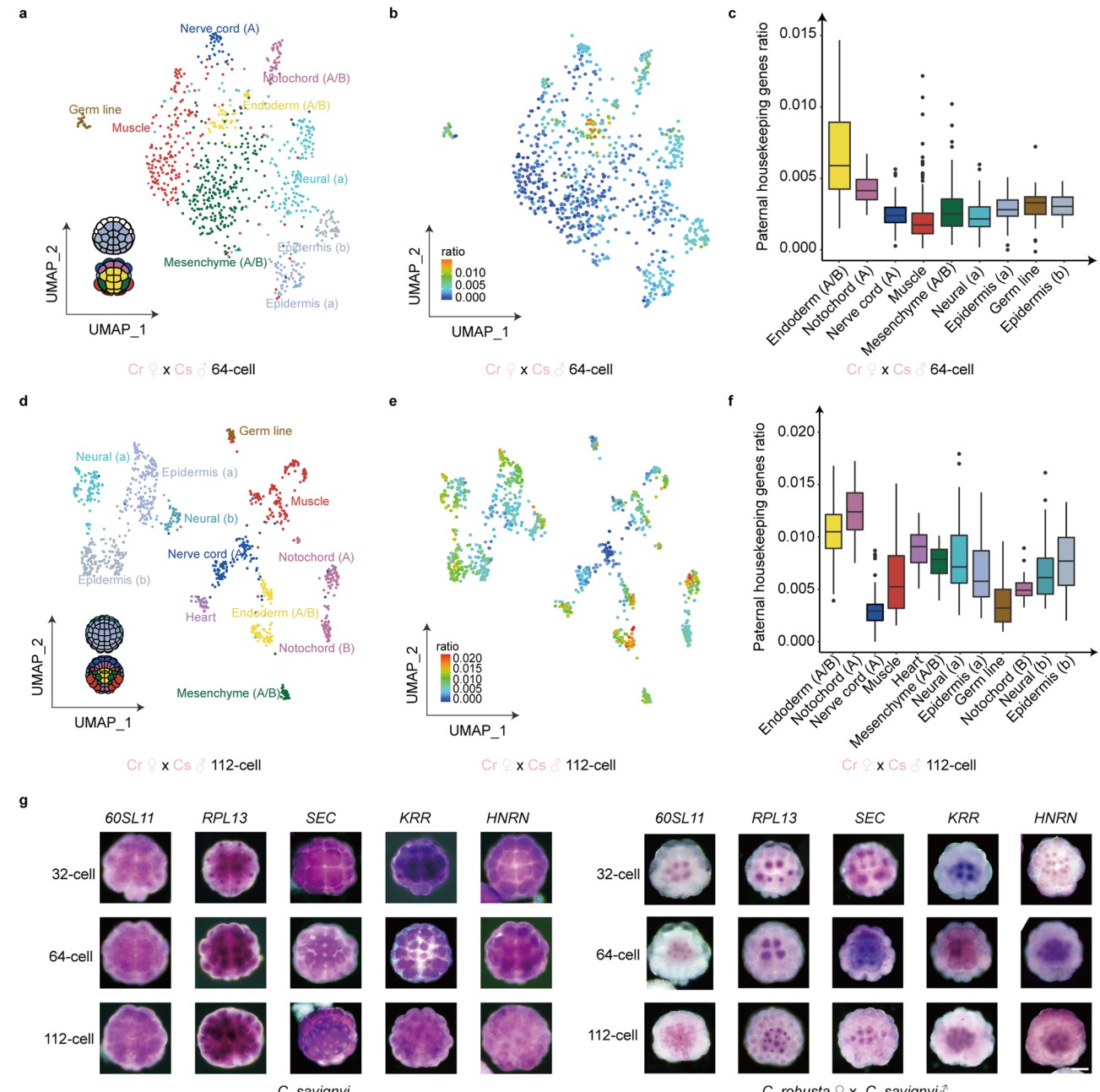

**Fig. 3 | Housekeeping gene reactivation revealed by single-cell transcriptome.** **a** UMAP plot of the 64-cell stage samples from forward cross embryos. In total, 844 cells were divided into 9 clusters and annotated as 8 cell types. The animal (up) and vegetal (down) blastomeres of a *Ciona* embryo at the 64-cell stage are shown. Font color corresponds to cell types. **b** Expression ratio of paternal housekeeping genes to all genes in each cell at the 64-cell stage from forward cross embryos. Colors represent the level of the expression ratio. **c** Boxplot of the expression ratio of paternal housekeeping genes in each cell type at the 64-cell stage from forward cross embryos. $n$ = 40, 43, 32, 187, 259, 150, 70, 20, 43 cells for each cell type from endoderm to epidermis (**b**). **d** UMAP plot of the 112-cell-stage samples from forward cross embryos. In total, 1002 cells were divided into 12 clusters and annotated as 9 cell types. Animal (up) and vegetal (down) blastomeres of a *Ciona* embryo at the 112-cell stage are shown. Font color corresponds to cell type. **e** Expression ratio of paternal housekeeping genes to all genes in each cell at the 112-cell stage from forward cross embryos. Colors represent the expression ratio. **f** Boxplot of the expression ratio of paternal housekeeping genes in each cell type at the 112-cell stage from forward cross embryos. $n$ = 98, 60, 107, 138, 33, 29, 95, 181, 27, 48, 43, 143 cells for each cell type from endoderm to epidermis (**b**). **g** Whole-mount in situ hybridization of *60SL11*, *RPL13*, *SEC*, *KRR* and *HNRN* in self-cross *C. savignyi* embryos and forward cross embryos at 32-, 64- and 112-cell stages. Scale bar: 50 μm. In each box plot, the horizontal black lines represent median values; boxes extend from 25th to 75th percentile of each group's distribution of values; the vertical extending lines indicate adjacent values, and dots mark observations outside the range of adjacent values. The experiment was independently performed at least three times. Source data are provided as a Source Data file.

housekeeping genes were asynchronized reactivated in hybrid embryos, and thus did not follow the maternal clock model.

To test whether the asynchronized reactivation of housekeeping genes followed the classic N/C ratio model, wherein the increasing cell cycle titrated maternally supplied repressors to relieve transcriptional repression, we collected the data of cell division and cell volume from previous data[32]. The results showed that the reactivation of housekeeping genes was neither fully correlated with the time after the last cell division in each cell lineage nor correlated with cell volume (Supplementary Fig. 16; Supplementary Data 10).

Given that endoderm invagination is the first event of gastrulation[33], the earlier major initiation of ZGA may be related to the cell mechanics, such as cell movements and cell shape changes. The stiffness of ascidian embryonic cells (Supplementary Data 11) has been investigated using atomic force microscopy[34]. Therefore, we compared the cell stiffness and paternal housekeeping gene expression in different cell types and found that earlier ZGA in endoderm and adjacent notochord cells was indeed associated with stiffness properties in the early developing ascidian embryos (Supplementary Fig. 17a). The expression profiles of stiffness-related signaling molecules, such as *Rho*, *Ephrin*, and *Nodal*[35,36], were also correlated with ZGA in different cell lineages (Supplementary Fig. 17b; Supplementary Data 12).

To address the possible mechanisms of the dynamics of ZGA, we investigated the relationship between cell fate and cell stiffness. We used U0126 to transfer the cell fate differentiation of notochord cells[37] and measured the stiffness using vinculin tension sensor based on fluorescence resonance energy transfer (FRET)[38,39]. A *Foxa.a* promoter-derived vinTS was expressed in these embryos to trace the cell stiffness measured in this fate-transferred notochord cell lineage[40]. Compared with the control group, the shapes of the *Foxa.a* promoter-expressed cells were significantly changed (Supplementary Fig. 17c). The mean FRET intensity in the *Foxa.a*-positive cell cortex in the U0126-treated group was significantly higher than that in the control group, which reflects lower cell stiffness (Supplementary Fig. 17c), suggesting a correlation between cell fates and cell stiffness.

Thus, we revealed the spatial heterogeneity of housekeeping gene reactivation using the unique *Ciona* hybrid system and single-cell transcriptome. Our results showed the ZGA timing was neither consistent with elapsed time nor with the N/C ratio model, but was determined by cell fate and might be correlated with cell stiffness.

## Species-biased allelic gene activation during ZGA

The sequence discrepancy between the parental gene orthologs also allowed us to examine the expression preference from either paternal or maternal, i.e., ASE. Among 5,879 orthologous gene pairs detected, 341 pairs showed zygotic-only expression in both forward and reverse cross embryos.

We divided these 341 gene pairs into four groups according to their expression ratio between *C. robusta* and *C. savignyi* in both hybrid directions. The early-expressed genes (from the 8-cell to 32-cell stage) (Supplementary Fig. 18a) tended to be *C. robusta*- or maternal-dominant, which could be attributed to that maternal-deposited TFs had advantage at recognizing enhancers from maternal DNA. During the transition stage (the 64-cell and 112-cell stage) (Supplementary Fig. 18b), more *C. savignyi*- and paternal-dominant genes were detected. Among the late-expressed genes (from the 112-cell to late-neurula stage), 138 and 106 were *C. robusta*- and *C. savignyi*-dominant, whereas 90 and 7 genes were maternal- and paternal-dominant, respectively, in both forward and reverse cross embryos (Fig. 4a, b, Supplementary Fig. 18c, d). These gene pairs showed the expression differences between allelic genes, especially from the 112-cell to late-neurula stage. Increased allelic differential expression in hybrid embryos tended to be species- rather than parental- manner during major ZGA. GO enrichment analysis of ASE genes revealed that species-biased genes were enriched in the immune system process and ion transport, whereas parent-biased genes were enriched in the metabolic process and regulation of transcription process (Supplementary Fig. 19).

To validate the biased expression features during zygotic genome activation, the CUT&Tag results were visualized on the biased-expressed genes. Although the signal pattern of H3K27ac was weak at early stages, the resulting curves and heatmaps still showed the relative signals for genes with expression-biased features (Supplementary Fig. 20). Under most circumstances, for *C. robusta* biased genes, signals mapped on *C. robusta* genome have higher peak in both results targeting H3K27ac and RNA pol II, implying a dominant

characteristic for transcription suites from *C. robusta*. For *C. savignyi* biased genes, signals mapped on *C. savignyi* genome showed an improved trend compared to that on *C. robusta* genome, which is consistent with the activation of respected genes. In maternal-based genes, a similar trend was also observed. The modification peaks from hybrid ascidians showed consistent trends with RNA-seq results (Supplementary Fig. 20), which also supported more species-biased allele-specific activation of gene expression.

We inferred that genes that showed species-biased expression in hybrid *Ciona* might be due to variations of cis-regulatory elements. Considering the sequence similarities were related to the conservation of cis-regulatory elements[41,42], we thus compared the sequence identity in the 1000-bp upstream of the initiation codon of CDS in different gene pairs between *C. robusta* and *C. savignyi* (based on the previous ATAC-seq data at 64-cell and 112-cell stage of *C. robusta*[43], 61.7% of the open chromatin regions are within 1 kb of initiator methionine among the peaks in promoter region). The sequence identity of upstream 100−300 bp was significantly lower in species-biased gene pairs than in parent-biased gene pairs or Benchmarking Universal Single-Copy Orthologs (BUSCO) gene pairs (Supplementary Fig. 21), suggesting the difference of cis-regulatory elements in this region in species-biased gene pairs.

Therefore, we analyzed the motif distribution of these allelic genes targeting at upstream 100−300 bp region (Supplementary Data 13). The biased motif distribution pattern was identified in ASE genes subject to species-of-origin effects (Supplementary Fig. 22). For example, the distribution of the predicted binding motifs of *FoxF* and *titin* in homolog genes in *Ciona* showed a *C. robusta*-dominant pattern, whereas predicted motifs regulating *IAP2* (inhibitor of apoptosis protein 2) and *ZicL* (Zic-like zinc finger transcription factor) showed a *C. savignyi*-dominant pattern. These findings inferred that the biased expression of gene pairs in species-of-origin effects may be attributable to the divergence of cis-regulatory elements between allelic genes. The H3K27ac and Pol II occupancy detected by CUT&Tag at the early neurula stage of *titin* in reverse cross embryos also showed that the signals aligned upstream of start codon in *C. robusta* exceeded that in *C. savignyi* (Fig. 4c).

To validate the different motif patterns were related to gene expressions, the promoter activities from both species were tested. Plasmids were constructed with the upstream promoter region of *titin* and *FoxF* from *C. robusta* and *C. savignyi*. By swapping the potential cis-regulatory element sequences of the upstream promoter region to the equivalent promoter region of the other species, two swapped vectors were also constructed (Supplementary Data 14). These plasmids were then transferred into *Ciona* embryos, respectively to observe their expression activities. The results revealed that embryos with the promoter from *C. robusta* showed signals at the early neurula stage (Fig. 4d), which is consistent with the results of motif analysis. The motif swap experiment of *titin* showed that *titin* expressed upon replacing the cis-regulatory element region from *C. robusta* into equivalent region of *C. savignyi* at the early neurula stage. The differences in promoter activities were also observed for *FoxF*, which also showed earlier expression with promoter from *C. robusta* compared to that from *C. savignyi* (Supplementary Fig. 23), suggesting that the species biased gene expression is associated with the differences in cis-regulatory element at the genomic level.

## Discussion

Herein, we developed a hybrid system using two deeply diverged ascidian species to symmetrically document the hierarchy and dynamics of ZGA. Natural hybrids of *C. robusta* and *C. savignyi* have not been reported because of the chorion membrane of the egg prevents cross-species fertilization[26]. However, enzymatic removal of this membrane enables hybrid embryos to be derived in both directions[26,44]. To the best of our knowledge, these are one of the most

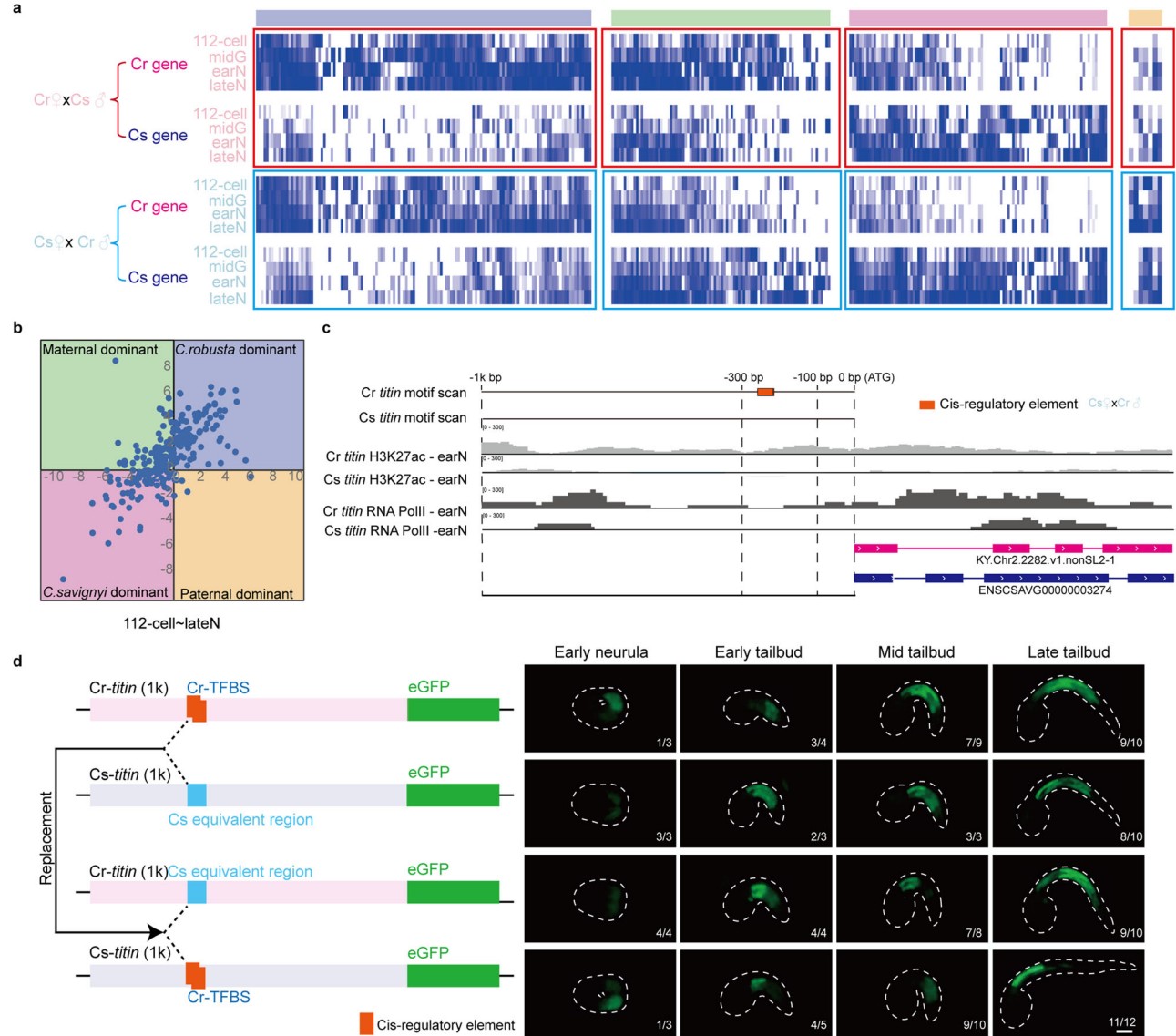

**Fig. 4 | Allelic gene activation during embryogenesis of hybrid animals.**
**a** Expression heatmap of zygotic-only expressed genes from the 112-cell to late-neurula stage from both directional crosses. The genes were divided into four groups according to the expression preference. The purple, green, pink and orange groups contained *C. robusta*-, maternal-, *C. savignyi*- and paternal-dominant genes, respectively. **b** Gene expression preference quadrant for zygotic-only expressed genes from 112-cell to late-neurula stage. The quadrant is divided into four parts. X axis indicates the $\log_2$ transformed expression ratio between genes from *C. robusta* and the allelic genes from *C. savignyi* in reverse cross embryos. Y axis indicates the $\log_2$ transformed expression ratio between genes from *C. robusta* and the allelic genes from *C. savignyi* in forward cross embryos. Therefore, the first and third quadrants indicate genes with *C. robusta*- or *C. savignyi*-dominant expression,

respectively, whereas the second and fourth quadrants indicate genes with maternal- or paternal-dominant expression, respectively. **c** Comparison of the distribution of a predicted cis-regulatory element and H3K27ac and Pol II occupancy by CUT&Tag at early neurula stage of *titin* gene from *C. robusta* and *C. savignyi*. The gene model and the predicted cis-regulatory element of *titin* were labeled. **d** Promoter activity test of *titin*. The promoters were cloned from *C. robusta* and *C. savignyi*, respectively. The swapped plasmids were cloned with exchange of motif regions. The plasmids were microinjected into *C. savignyi* embryos, and the GFP signals were observed from the early neurula to late tailbud stage. Scale bar: 50 μm. The experiment was independently performed at least three times. Source data are provided as a Source Data file.

distinct hybrid embryos obtained in multicellular organisms[45,46]. Therefore, the hybrids provide a unique system for the study of ZGA dynamics, thereby helping overcome the challenge of distinguishing zygotic transcripts from maternally deposited RNAs. Additionally, studying on embryogenesis of distant hybridization using hybrid ascidians will help us understand the evolution of genome incompatibility. The sequence discrepancy between the two parental genomes also enabled the examination of the ASE during ZGA.

We used RNA-seq and CUT&Tag for distinct ascidian hybrids to detect ZGA temporal dynamics and revealed two coordinated waves of ZGA especially from the 16- to 32-cell stage and from the 64- to 112-cell

stage, along with housekeeping gene reactivation. Thereafter, we identified a different ZGA pattern, wherein the housekeeping gene reactivation could neither be explained by absolute timing nor by the N/C ratio.

In addition to the variation in cell size, another explanation has been proposed for the different timing of ZGA in *Xenopus* embryos. This explanation concerns the expression timing of specific germ-layer genes, indicating that the animal pole-derived ectoderm is the earliest lineage to activate the genome[47]. Because housekeeping gene reactivation is the hallmark of major ZGA, the fact that the earliest reactivation of housekeeping genes occurs in vegetal lineages infers that

ZGA timing in *Ciona* was not correlated with germ layer but with cell fate determination, which is different from ZGA timing in *Xenopus*.

A study of the temporospatial dynamics of single-cell stiffness in the ascidian embryo revealed greater stiffness in the endoderm and adjacent notochord cells[34]. With the data, we found that ZGA timing was correlated with the cell stiffness properties and also detected the decreased cell stiffness under transition of notochord cells. A large variation in cell mechanical properties has been reported at the single-cell level[48]. The differential cell surface fluctuations account for the lineage sorting of the primitive endoderm from the epiblast[49], as the primitive endoderm displays higher surface fluctuations than the epiblast. These nonequilibrium cell surface dynamics could be an outcome of the asynchronized ZGA of cellular function genes in *Ciona*. Moreover, the housekeeping gene reactivation also could trigger the change in cell behavior, which is crucial for the subsequent organogenesis.

Thus, we propose another possible model for ZGA dynamics based on our findings that the ZGA timing was neither consistent with elapsed time nor with the N/C ratio model, but was determined by cell fate and correlated with cell stiffness.

Using the powerful hybrid ascidian system, we found that species-rather than parent-of-origin ASE effects occurred during ascidian ZGA. The allelic gene activation during embryogenesis of hybrid *Ciona* showed that 72% genes exhibited species-of-origin effects, whereas the remaining 28% genes exhibited parent-of-origin effects. These results are comparable with other mammals reported, wherein many ASE genes in humans were associated with the immune response[50], whereas the parent-of-origin effects were enriched in metabolic traits in mice[51]. Evidence from large-scale studies in mammals also suggested that the majority of the imbalance of ASE resulted from genetic effects rather than imprinting or random monoallelic expression[52]. Moreover, we theoretically and experimentally verified that the mechanism of the species-of-origin ASE effect might be attributed to the divergence of cis-regulatory elements between the two species. A complex interplay occurs between TFs and DNA binding motifs across animals and is considered as a key driver of phenotypic variation[53,54]. The effects of sequence variation of cis-regulatory elements on differential allelic TF occupancy have been revealed in mammals[55,56]. Therefore, the variation in the cis-regulatory landscape might also contribute to the divergence of ZGA across species. In addition, considering the trans-splicing events in *Ciona*[57], the cis-regulatory elements may thus be located distance away from the upstream 100–300 bps. The scan of open chromatin regions within and upstream of the genes will help to comprehensively uncover the active regulatory elements.

In summary, by taking advantage of the natural "genetic tags" from distinct *Ciona* hybrids, we revealed two waves of ZGA, from the 16- to 32-cell stage and from the 64- to 112-cell stage, along with housekeeping gene reactivation. Intriguingly, we identified a different ZGA pattern, wherein the housekeeping gene reactivation could neither be explained by absolute timing nor the N/C ratio but showed a significant correlation with cell fate. Species- rather than parent-of-origin ASE effects occurred during ascidian ZGA. Furthermore, we verified that the divergence of cis-regulatory elements of two species could be the mechanism for ASE effects.

These findings propose a possible model for understanding the potential mechanisms of ZGA dynamics and support the notion that ZGA, especially the reactivation of housekeeping genes, is under extreme selection pressure. Considering that *Ciona* is the closest invertebrate relative of vertebrates, our study also provides a system for understanding the evolution and adaptation of strategies that regulate this critical developmental process.

## Methods

### Animal hybrids and sample collection
Adults of *C. robusta* and *C. savignyi* were collected in Weihai, China, and bred in seawater at 18 °C in the laboratory. Eggs and sperm were obtained from mature animals as described previously[27]. Embryos from *C. savigyni* sperm fertilization of *C. robusta* eggs were defined as forward cross while embryos from *C. robusta* sperm fertilization of *C. savignyi* eggs were defined as reverse cross. After 10 min, the fertilized eggs were washed twice with filtered seawater. Embryos were raised to different stages at 18 °C[58]. Forward and reverse cross embryos at the 1-, 2-, 4-, 8-, 16-, 32-, 64-, and 112-cell; mid-gastrula; and early- and late-neurula stages were used for morphological observation and sequencing.

### Bulk RNA-seq and data analysis
For each sample, 100 – 500 morphologically normal embryos were randomly selected and transferred to tubes precoated with 0.1% bovine serum albumin (BSA). Total RNA was extracted using TRIzol RNA extraction reagent. The Nanodrop spectrophotometer and Agilent 2100 bioanalyzer were used to assess RNA integrity. Transcriptome sequencing of RNA sample was performed by Novogene (China) using an Illumina NovaSeq system in paired-end 150-bp mode. Reads were aligned to the *C. robusta* (KY2019; Ghost, http://ghost.zool.kyoto-u.ac.jp/default.html) and *C. savignyi* (CSAV2.0; Ensembl, https://asia.ensembl.org/index.html) genomes using HISAT2 software v2.0.5[59]. FeatureCounts (1.5.0-p3)[60] were used to acquire the reads counts mapped to each gene and then to acquire the FPKM value. FPKM of each gene was then calculated based on the length of the gene and reads count mapped to this gene[61]. A co-expression gene network for transcriptomic datasets was constructed using the R package WGCNA[62].

### Bulk CUT&Tag experiment and data analysis
Embryos at 16-, 32-, 64-, and 112-cell and early neurula stages of reverse cross were collected for construction of CUT&Tag libraries of both H3K27ac and RNA Pol II at different stages. CUT&Tag was performed on the pre-washed nucleus with anti-RNA polymerase II antibody and anti-H3k27ac antibody as described[63]. The raw data were split with python tool barcode splitter (version 0.18.6) according to sequencing barcodes. The fastq files were aligned against a combined reference genome of *C. robusta* and *C. savignyi* using Bowtie2 version 2.4.4[64]. Bam files were transferred to browsable BW file using bamCoverage version 3.5.1. The computeMatrix (version: 3.5.1) scale-regions function was used to compute the signal. The plotHeatmap was used to visualize the results with respected BED files of genes. Genome browser snapshot was captured with Integrative Genomics Viewer (IGV, v2.12.2)[65] using Run Batch Script tool. DeepTools (v3.5.1)[66] was used to plot peak distribution.

### Cell dissociation of embryos
Morphologically normal embryos at the 64- and 112-cell, and early neural stages were collected and transferred to tubes precoated with 5% BSA in $Ca_2^+$-free artificial sea water (ASW). Embryos were immediately dissociated using 0.2% trypsin in $Ca_2^+$-free ASW with 5 mM EGTA, after which they were pipetted for 3 min on ice to complete their dissociation of embryos into individual cells. Digestion was inhibited by adding 0.2% BSA in $Ca^{2+}$-free ASW. Cells were then collected via centrifugation at 4 °C and 500 g for 2–5 min and resuspended in ice-cold $Ca^{2+}$-free ASW containing 0.1% BSA.

### Single-cell RNA-seq
For 64- and 112-cell stage embryos from forward cross, single-cell gel beads in emulsion were generated using a Chromium Controller instrument (10× Genomics). Sequencing libraries were prepared using Chromium Single-Cell 3′ Reagent Kits (10× Genomics) according to the manufacturer's instructions. After performing cleanup using a SPRIselect Reagent Kit, the libraries were constructed by performing the following steps: fragmentation, end-repair, A-tailing, SPRIselect cleanup, adaptor ligation, SPRIselect cleanup, sample index PCR, and

SPRIselect size selection. For early neurula-stage embryos from forward cross, sample preparation, library construction, and single-cell sequencing were performed using a BD Rhapsody Single-Cell Analysis, for which cell capture beads were prepared and loaded onto the cartridge. Cartridges were washed, cells were lysed, and cell capture beads were retrieved and washed prior to performing reverse transcription and treatment with Exonuclease I. cDNA libraries were prepared using mRNA-targeted sample tags, and a BD AbSeq library was prepared using the BD Rhapsody-targeted mRNA and AbSeq Amplification Kits. For 112-cell stage embryos from reverse cross, library was performed using single cell 3' protocol (BMKMANU DG1000). Libraries were then sequenced using an Illumina NovaSeq system.

## Single-cell RNA-seq data analysis

For 10× Genomics data, bcl files were converted into FASTQ format using bcl2fastq v1.8.4 (Illumina). To generate gene–barcode matrices for each sample, Cell Ranger v3.0.2 were used with its default settings. For BD Rhapsody data, the Seven Bridges platforms pipe line v1.9 (https://www.sevenbridges.com) was used. For BMK data, the matrix file was acquired from Biomarker technology company (China). We mapped sequencing reads to the reference genome combining both *C. robusta* and *C. savignyi*. To remove signals from hypothetical empty droplets or degraded RNA, a preliminary filter was performed to preserve all genes expressed in at least three cells as well as at least 250 genes expressed in a single-cell. We normalized the read counts of each cell using Seurat V4[67], and 2000 highly variable genes were selected from each sample for downstream principal component analysis. For forward cross embryos, the 64-cell stage sample was filtered using 1000–7000 detected genes; the 112-cell stage sample was filtered using 2000-7000 detected genes; the early neurula-stage sample was filtered using 500-4500 detected genes. For reverse cross embryos, the 112-cell stage sample was filtered using 700-4500 detected genes; Cell distance was visualized using the UMAP method in reduced two-dimensional space. The cell clusters were annotated according to the marker genes in each cell cluster. The Student's t-test was used for the statistics analysis of paternal transcript ratio in different cell types.

## In situ hybridization

In situ hybridization was performed at the 32-, 64- and 112-cell stages in *C. savignyi* embryos and forward cross embryos. Digoxigenin (DIG)-RNA probes were synthesized by in vitro transcription with T7 and SP6 RNA polymerase. Embryos were fixed with 4% paraformaldehyde (PFA) in seawater at 4 °C for overnight. Embryos were washed with phosphate-buffered saline containing 0.1% Tween 20 (PBST), and then treated with 10 µg/mL proteinase K for 30 min at 37 °C, then washed again with PBST and fixed with 4% PFA for 0.5 h at room temperature. Embryos were then incubated in 20× saline sodium citrate buffer (SSC), 50% formamide, 5× Denhardt's solution and 0.01% Tween 20 for 10 min at room temperature. Embryos with 1 ng/ul probe were then incubated in 0.5× SSC, 50% formamide, 100 µg/mL yeast tRNA, heparin and 0.1% Tween 20 for 20 h. Embryos were then incubated with 20× SSC, 5× Denhardt's solution and 0.01% Tween 20 for 15 min four times, following which they were incubated in 1% blocking reagent (Roche) in PBST for 20 h at 4 °C and then washed with NBT and BCIP for detection. The probe primers used in this study are listed in Supplementary Data 15.

## Inhibitor treatment and cell stiffness measurement

The *Foxa.a* promoter was amplified from genomic DNA of *Ciona* (primers are listed in Supplementary Data 15). The PCR products were subcloned at the upstream of vinTS and Lck::mScarlet to construct a notochord traced FRET tension sensor and membrane marker. Then the Foxa.a>vinTS and Foxa.a>Lck::mScarlet plasmid was co-electroporated into the dechorionated eggs. Fertilized eggs

were cultured at 14°C until the 16-cell stage. U0126 (HY-12031A; MCE) was dissolved in DMSO as a 10 mM stock. A total of 100 tension sensor electroporated 16-cell embryos were moved to a 6-well plate with 5 mL filtered sea water. U0126 stock (1 µL; 1 µL DMSO in control group) was added to a final concentration of 2 µM[68]. Embryos were cultured at 14 °C. The expression of the vinTS was traced each hour by using fluorescence stereomicroscope until the fluorescent intensity is sufficient for imaging at middle tailbud stage. The FRET imaging was conducted using an Olympus line scanning confocal microscope equipped with 445 nm emission laser. The mTFP and mVenus signals were detected coinstantaneous with the same gain; the Lck signal was then collected. The FRET intensity distribution image was calculated with ImageJ (Image Calculator) using the formula mVenus/mTFP. Then, a 10-pixel wide freehand line region of interests along the Lck signal was set per cell and overlaid on the FRET intensity distribution image to measure the mean FRET intensity.

## Identity comparison of gene upstream sequence

For species-biased gene and parental-biased gene sets, 244 gene pairs (*C. robusta*-dominant and *C. savignyi*-dominant genes) and 97 gene pairs (maternal- and paternal-dominant genes) were used for identity comparison. The 1000 bp upstream regulatory sequences of homolog genes from *C. robusta* and *C. savignyi* were extracted. The upstream sequence was separated into sequencing fragments with 100 bp lengths. For each homolog gene pair, the identity of regulatory sequence in the same location was calculated using EMBOSS needle localized script. One-tail Wilcon test was conducted between the identity value of BUSCO and species- and parental-biased gene sets. The identity value was visualized with beeswarm plot. The *P*-value result was visualized with broken line plot.

## Motif scanning for species-biased genes

Motif distribution scanning was conducted using FIMO in the MEME suite[69]. Candidate motif frequency matrices were obtained from urochordata CORE dataset from the JASPAR database (https://jaspar.genereg.net/)[70]. All 86 publicly available motif profiles were used for motif scanning. The sequence database was constructed with the 100–300 bp upstream sequences of all 244 pairs of species-biased genes. The output threshold (hits with a *P*-value less than the threshold) was set to 0.0001.

## Molecular cloning, plasmid construction and microinjection

The upstream regions of *titin* and *FoxF* from *C. robusta* and *C. savignyi* were PCR-amplified (gene IDs and primers are listed in Supplementary Data 15) from genomic DNA and cloned into an expression vector containing EGFP downstream using Xho1 and Kpn1 restriction enzyme (Thermo), respectively. Cis regulatory element regions and equivalent regions were swapped by PCR-amplification. Plasmid purification was performed using EndoFree Mini Plasmid Kit and were diluted to 40 ng/µL for microinjection. Mature eggs were isolated from the adult *Ciona* and then dechorionated before microinjection. Plasmids were microinjected[71] into the dechorionated eggs using the Eppendorf transferman microinjection system. After injection, the eggs were fertilized in vitro and cultured at 18 °C until the tailbud stage. Then, the fluorescent signals were recorded with Zeiss 980 laser confocal microscope.

## Statistics and reproducibility

No statistical method was used to predetermine sample size. No data were excluded from the analyses. The two-sided unpaired *t*-test was used for statistical evaluation of two-group comparisons. The non-parametric Wilcoxon rank sum test was utilized for marker gene identification analysis. The adjusted *p*-value is based on Bonferroni correction using all features in the dataset.

## Reporting summary

Further information on research design is available in the Nature Portfolio Reporting Summary linked to this article.

## Data availability

All data generated or analyzed during this study are included in the manuscript and supporting files. The bulk RNA-sequencing, single-cell RNA-sequencing and CUT&Tag sequencing datasets generated in this study have been deposited at NCBI under the accession numbers PRJNA819937 PRJNA820327 PRJNA1003395 and PRJNA1002800. The databases used in this study include JASPAR. Source data are provided with this paper.

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

## Acknowledgements

This work was supported by the National Key Research and Development Program of China (2022YFC2601302, to B.D.), the Science & Technology Innovation Project of Laoshan Laboratory (No. LSKJ202203002, to B.D.), the National Natural Science Foundation of China (Grant No. 32270560, to K.C.), and the Taishan Scholar Program of Shandong Province, China (to B.D. & S.W.). K.C. would like to thank Hao Wang for insightful discussion.

## Author contributions

B.D. and K.C. conceived and designed the experiments. J.K.W. and W.Z. conducted most of the experiments (with help from H. P., Q.Z., J. B., Y.G. and L. Z.) L.Li., A.J., Y.L. and J.W. with help from H.Y., L.Leng., L.W., P.L. and S.W. performed the data analysis. Initial draft was drafted by J. W. and W. Z. The final version of the manuscript was prepared by B.D., K.C. and L.Leng. All authors approved the manuscript.

## Competing interests

The authors declare no competing interests.
