## [Peer Review File · Nature Communications]

Temporospatial hierarchy and allele-specific expression of zygotic genome activation revealed by distant interspecific urochordate hybridsReviewer #1 (Remarks to the Author):

In this manuscript, Dong and colleagues present a very astute and powerful model system to characterize the maternal to zygotic activation in ascidian embryos: interspecific hybrids between two distantly-related *Ciona* species. They show that these hybrids develop to the tailbud stages and that it is possible to allele-specific dynamics of expression of each gene according to its parent or species of origin.

The main strength of the article is the hybrid system used, which is strikingly powerful. The reported experiments analyses are overall sound. Unfortunately, the authors try to cover too much ground, reducing the depth of each analysis. As a consequence the findings reported are either of a confirmatory nature or insufficiently developed.

Major suggestions :

This article reads like a test run for more interesting studies using *Cirobu/Cisavi* hybrids. It reports many interesting successive snippets of information, each worthy of a more in-depth study. Yet the manuscript currently fails to integrate these snippets into a convincing story. It also lacks a remarkable unexpected result that would make it suitable for a major journal. My main advise would be to choose one of these snippets and dig deeper in the hope to find something unexpected. One possibility would be to analyse more deeply the differences in genetic programs between the two species.

Some parts are difficult to follow because of vague or insufficient explanations. This applies in particular to the analysis of the regulatory relationships between maternal and successive waves of zygotic genes (lines 171 to 187). After reading the main text and the materials and methods, I could still not really understand what had been done and whether it made any sense. I was not even sure that the « motif scanning » section was relevant as the 244 genes described here do not seem to match those of lines 171 and following. Providing a list of the genes being analysed would be a first step. I was also surprised that the authors searched for motifs in the 100-300 bases immediately upstream of what I understood to be the initiator methionine. Many *Ciona* genes are trans-spliced with 5' outrons. The TSS may thus be located some distance away from the initiator methionine (see Matsumoto et al., *Genome research*, 2010). It would be much preferable to scan open chromatin regions within and upstream of the genes of interest (see Madgwick et al., *Dev. Biol.* 2019).

The results are frequently stated but not interpreted. It is therefore difficult to tell apart mere confirmatory results from true interesting novelties. For instance, do the authors consider the discovery that different classes of genes have different « ages » according to phylostratigraphy as a major result or as a confirmation in *Ciona* of what has been found in other systems?

Are the genes composing the modules reported line 150 and following orthologous between species ? or do the modules consist in different genes in the two species ?

Lines 232-23 : I am not convinced of the claimed lack of correlation between the timing of activation of housekeeping genes and the time elapsed since the last cell division. For instance at the 64-cell stage these genes are preferentially expressed in endoderm and notochord (Fig. 3g) and the precursor cells in these two tissues are the ones that have been around for the longest time (since the 44-cell stage).

Minor comments :

Important concepts should be better defined. For instance what is MBT precisely ? If it is defined by the desynchronisation of cell divisions, then this occurs in *Ciona* just after the 16-cell stage (see Nishida's early work). Also the 2 « waves » of zygotic transcription that define ZGA should be better defined, and called the same way throughout the ms (currently they are referred to as first and second waves, and as minor and major waves). Also how do the authors define « housekeeping » genes ?

Line 92 : *Ciona savignyi* is not « distributed across most coasts around the world » : WORMS indicates that it is mostly found in the North Pacific ocean.

While *Halocynthia roretzi* counts 110-cells at the onset of gastrulation, *Ciona* and other phlebobranch species count 112-cells at the same stage (due to the precocious division of the A7.6 cell pair ; see Tassy et al. *Curr. Biol.* 2006 ; Guignard et al. 2020).

Lines 74 and 197 : The idea that ascidians develop using a mosaic strategy (i.e. one in which cells do not communicate with each other) has been shown wrong by many authors (starting with H. Nishida). The authors are right that ascidian embryos develop with invariant lineages (Conklin, Satoh, Nishida, Christiaen, Lemaire labs among many others), but this is different from « mosaic » development. The term mosaic should be avoided in this context.

Lines 248 : Why do the authors describe Nodal and Ephrin genes as « stiffness genes » ? Is there a reference for this ?

Lines 289 and Figure 4d : I do not understand why the authors only show binding sites for Fox family genes ?

Lines 226-227 : I do not understand this sentence.

Lines 263-264 : I do not understand this sentence.

Line 363 : what does « alone » mean ?

Reviewer #2 (Remarks to the Author):

SUMMARY

In this study, Wei et al. used forced hybrids between two ascidian species, *Ciona robusta* and *Ciona savignyi*, which have widely divergent genomes, to profile the transcriptomes of embryos at successive early stages, and identify parent specific gene expression, to gain insight into the genome-wide features of zygotic gene activation (ZGA) in ascidians.

Focusing on paternal vs. maternal gene expression of housekeeping genes, the authors propose that ZGA occurs in two waves. Using single cell RNA-seq, and chromogenic in situ hybridization, the authors uncover lineage and spatial differences in the relative proportion of paternal-genome-derived housekeeping gene expression, which they relate to asynchronous ZGA. The authors propose that neither the nucleocytoplasmic ratio nor the timing after cell division account for these spatiotemporal differences, but cell stiffness.

Finally, the authors reveal species-specific differences in allelic gene expression presumably attributable to differences in cis-regulatory sequences.

KEY MERITS AND GENERAL EVALUATION

Key merits of the study include the generally important problem of ZGA in tunicates, which remains understudied, as is the notion of asynchronous and lineage-specific zygotic genome activation. The authors decidedly leverage a unique and powerful biological feature, namely the ability to generate viable first-generation hybrids between species with very divergent genomes but very conserved early embryogenesis. The authors also decisively used modern transcriptomics approaches, including from single cells. The datasets generated are of potential high value, and the notion that ZGA occurs asynchronously in the various lineages of early embryos is well supported by the single cell RNA-seq and in situ data.

Nevertheless, there are many marked shortcomings throughout the study, which strongly reduced enthusiasm for this manuscript.

CONCERNS

- One of the key conclusions of the paper, that there are two waves of ZGA, is not convincingly supported by the data and analysis. That conclusion appears rather abruptly at the end of page 6, relying on plots shown in Ext Data Fig S2, which show complex dendrograms without much explanation. Moreover, the heatmaps shown on Fig. 2d, e could as well support the notion that ZGA is quite continuous, especially considering that this is bulk RNA-seq and the following section emphasize lineage specific timing.
- Fig. 2h is not supported at all and should be removed. There is no analysis of maternal gene product degradation in the paper.
- The metrics used to evaluate zygotic genome activation, namely the proportion of paternal RNA for each gene is flawed in several ways: it assumes, which may be largely true, that the sperm does not contribute to adding paternal RNA to the zygote. Also, because it is relative the total amount of RNA for each gene, including maternal RNA that may be degraded, the metric convolves both RNA synthesis, for each allele, and RNA degradation, from each genotype. This reviewer appreciates the attempt to get at genome activity with CUT&TAG for RNAPII and histone marks, but this is not primarily used. An unexploited option is to leverage intronic reads to more accurately measure RNA synthesis from each parental genome.
- The attempt to explain spatial/lineage differences in proportion of paternal transcripts is very tentative. Specifically, Ext Data Fig 16 does not mention ZGA at all, and there is no attempt to perturb either of the parameters (N/C ratio, timing after division, stiffness) and assay ZGA to try and establish a causal relationship. The conclusion that stiffness of endoderm progenitor cells somehow promote activation of the zygotic genome is thus not supported by the evidence presented.
- The last part connecting allele-specific zygotic gene expression to differences in cis-regulatory sequences is potentially interesting, but suffers from several caveats as well:
 - o the point is almost trivial and supported by studies in other systems; including the chosen cis-regulatory DNAs that have been studied in *Ciona* and show different results.
 - o the in silico cis-regulatory analysis is not convincing as it focuses on very short upstream regions, ignoring introns, more distal elements and generally the accessibility data available for *Ciona*; the motif used are tentative and it is not clear to this reviewer how the authors can effectively distinguish between all the Fox motifs (Ext Data Fig 20); finally the conclusion is not tested by changing candidate binding sites in cross-species reporter assays.
- Ultimately, the paper gives the impression that it attempts to include too many points without strong or informative support for the main conclusions. Specifically, the section on the presumed evolutionary ages of genes is potentially interesting but under-developed and not validated; Est Data Figs. 8-10 are not particularly informative. The study would strongly benefit from focusing on the key points and strengthening the experimental and analytical support.

Reviewer #3 (Remarks to the Author):

In this manuscript by J. Wei, W. Zhang, L. Leng, A. Jiang and colleagues, the authors are undertaking on a long-standing question regarding the critical stage of ZGA during metazoan embryogenesis, using an elegant hybrid model of ascidian species. The authors were able to demonstrate the minor and major ZGA waves in early development of the hybrid embryos using mostly gene expression analysis approaches. Moreover, the authors show that, in ascidian development, the spatial gene expression heterogeneity during ZGA is correlated to cell fate and cell stiffness rather than previously offered models. The authors also provide some evidence for regulatory connection between maternal and zygotic genes as well as ASE in a species, rather than parental, -related manner. The authors' findings linking ZGA timing to cell fate and stiffness, as well as, the discovery of species-biased allelic gene activation during ZGA, are of high interest to the field.

Although the gene expression analysis approaches in this manuscript hold a potential to unravel interesting evolutionary insights into ZGA regulatory mechanisms, overall, the manuscript lacks a deeper analysis into the claimed mechanisms as well as experimental evidences to support some of the statements and interpretations that are made in the text body. As the vast majority of the claims in the presented manuscript are based on RNA-seq data, without more detailed analysis and

further experimental follow-up, I suggest that the authors should rephrase and tone down some of them, and widely edit the figures to make them easier to follow.

While hybrid Ciona is presented as a very intriguing model to study ZGA and ASE by being the most distant hybrid embryos to date, it is only used as a descriptive system at this stage. It is unclear and necessary to explain with more data to the general reader how the early embryogenesis is proven to be intact in the hybrid Ciona given that the defects of the embryos start very early in organogenesis. A comprehensive and comparative characterization of early embryogenesis is needed between hybrid and non-hybrid embryos to demonstrate that the ZGA in hybrid embryos are intact and physiologically relevant. Furthermore, at least a proof-of-concept experiment that targets specific gene expression to intervene ZGA dynamics, cell fate or stiffness is necessary to confirm the causality in the interpretation of the results, apply the hybrid chordate system more widely and claim it as a novel system to study ZGA dynamics.

To further address some of the concerns, more substantial analyses are needed, such as detailed comparison of genes in different stages or waves, use of scRNA-seq data from both $Cr\sigma \times Cs\varphi$ and $Cs\sigma \times Cr\varphi$ hybrid crosses, validation of species-biased allelic gene activation results with additional methods such as H3K27ac and RNA PolII ChIP-seq, and demonstration of the species-dominant TFs with CUT&RUN data. Overall, the authors' work would benefit from a more in-depth analysis to strengthen their findings and interpretations.

Major points:

1. Line 99-100 - What does "slightly higher development failure rate" mean? In Figure 1c, tail bud stage looks quite different between reciprocal crosses. Could the authors provide more evidence than morphology by brightfield microscopy to claim "no significant difference in embryogenesis". Subsequently, can the authors provide quantification and/or description of the developmental failure? Could the authors provide more pictures as Extended Data? The statements are difficult to judge based on a single picture at each stage without replicates and more details on Ciona embryogenesis.
2. Line 101 - 'which is distant from our research window' - The research window should be introduced for ZGA in Ciona, therefore, either letting the reader to judge for themselves or guide the reader whether it is distant enough.
3. Figure 1d is not mentioned in the text.
4. Figure 2a,b,c are confusing and hard to follow. Does each color represent each reciprocal hybrid or each genome? The figure legend also does not give enough explanation. The red arrows seem to depict each ZGA stage but this should be written in the Figure legend.
5. Line 114 - How is the paternal ratio for both crosses at stage 16-cell is 0.02% when the plot in Figure 2a appears to have much more transcripts in one cross compared to the other? Overall, it seems that the $Cr\sigma$ expression is always higher, given the 'higher developmental failure rate' mentioned in Figure 1, I think that this should be further explored.
6. Line 121 - Is it possible for the maternal genome to have a stable expression bias over the paternal genome in any of these species or hybrids? Is it safe to interpret that "maternally inherited transcripts might still be retained until the neurula stage" without having data in the further stages?
7. In Figure 2c, blue embryos show higher de novo gene numbers for most timepoints. Does this mean that, in total, do they express more number of genes? If so, are these additionally scored genes lowly expressed genes? If so is the discrepancy due to thresholding or any difference in pooled sample size or depth of the RNA-seq?
8. Line 137 - 'Combining the appearance of gene and expression characteristics' - what does it mean? Some experimental details will help the reader.
9. The aim and presentation of results in supplemental figure 3 are unclear. Please provide clustering of the CUT&Tag peaks (to show what those few genes that are indeed bound are). Is PolII more enriched in the gene body compared to TSS? The tracks provided are without scales in panel b, making it impossible to assess the enrichment of the peaks. How many replicates were used? Why was only one hybrid and one stage tested?
10. Line 156 - "regulation" should be detailed in the text. What is it referring to?
11. Line 158 - In Figure 2d-e, what are the genes in each module and how much overlap are there for each module between reciprocal hybrids?

12. It appears that Fig 2g is not mentioned in the text at all. Additionally, in Fig 2g, the definitions for minor and major waves are confusing as they refer to the first and second wave, respectively. Moreover, the genes from the first and second wave are only defined in Fig 2 and not used in the following analysis. Can the authors clarify if genes from first and second wave show consistent results regarding the relation to cell stiffness and species-biased ASE?

13. In Figure 3, it is unclear which hybrid cross is used for scRNA-seq, $Cr\sigma \times Cs\varphi$ or $Cs\sigma \times Cr\varphi$? The scRNA-seq of the both hybrid crosses is needed to validate all the results in Fig 3. While the correlation of cell stiffness to ZGA timing is an interesting topic, the analysis presented is relatively shallow. For example, it may be more appropriate to use a scatterplot with R-value and P-value to show the correlation rather than a boxplot (Fig. 3h). Additionally, what happened to the 64 single cell data, and do they show similar results?

14. Figure 4a is difficult to read, and a boxplot may better show the pattern of the four groups. It is not clear whether the 341 pairs used in Fig 4a are identified by including all stages or only late stages (second wave?). Furthermore, a more comprehensive analysis of species-biased ASE should be done for different stages or for first and second wave genes. Are species-biased ASEs consistent in different stages or stage-specific?

15. In Figure 4d, It is unclear whether it is plotted for a specific gene or is just a schematic diagram. The authors claim that "the distribution of the trans-regulating element binding motifs of FoxF and titin in homolog genes in *Ciona* showed a *C. robusta*-dominant pattern, whereas motifs regulating IAP2 and ZicL showed a *C. savignyi*-dominant pattern." However, it is unclear whether any motif enrichment analysis was performed, and it is challenging to claim "dominant" without performing any ratio calculation or statistical tests. A more systematic analysis is needed to make such strong claims, such as performing CUT&TAG of FoxF, titin, IAP2, and ZicL to verify whether they truly bind to those genes in an allele-specific manner.

16. The authors could also use H3K27ac and RNA PolII data to validate the species-biased ASE findings.

17. Line 550 - In the methods, what is the 'Statistics analysis' referring to? It seems that the manuscript contains much more elaborate statistics.

Minor points:

1. The authors should use either "massive" ZGA or "major" ZGA throughout the manuscript for consistency.
2. Line 128 - All genes should be presented by their full name, when introduced for the first time in the text.
3. Line 154 - It should be "gene" instead of "gen".
4. Line 169 - "The specificity of ZGA might be related to the hourglass model of developmental evolution" - this is not a result. I will suggest leaving it to the discussion.
5. It would be helpful if the authors could label the hybrid cross and developmental stage in all figures (i.e., Fig3, Fig 4d,e, Extended Fig 3b).
6. Line 210 - it should be "in each cell type" instead of "in each cell" .
7. Line 220, it should be "validate" instead of "valid".
8. Line 227 - 'the developmental clock model' - This is the second time that this appears. The authors should address this in the introduction.
9. Lines 235-238 - a very long and unclear sentence. Please rephrase.
10. Overall, the methods section should be revised and edited on the grammar level.

A point-by-point response

Ref: NCOMMS-23-03866

Title: Temporospatial hierarchy, dynamics and allele-specific expression of zygotic genome activation in basal chordates revealed by distant interspecific hybrids

REVIEWER COMMENTS

Reviewer #1 (Remarks to the Author)

In this manuscript, Dong and colleagues present a very astute and powerful model system to characterize the maternal to zygotic activation in ascidian embryos: interspecific hybrids between two distantly-related *Ciona* species. They show that these hybrids develop to the tailbud stages and that it is possible to allele-specific dynamics of expression of each gene according to its parent or species of origin. The main strength of the article is the hybrid system used, which is strikingly powerful. The reported experiments analyses are overall sound. Unfortunately, the authors try to cover too much ground, reducing the depth of each analysis. As a consequence, the findings reported are either of a confirmatory nature or insufficiently developed.

Major suggestions:

This article reads like a test run for more interesting studies using *Cirobu/Cisavi* hybrids. It reports many interesting successive snippets of information, each worthy of a more in-depth study. Yet the manuscript currently fails to integrate these snippets into a convincing story. It also lacks a remarkable unexpected result that would make it suitable for a major journal. My main advise would be to choose one of these snippets and dig deeper in the hope to find something unexpected. One possibility would be to analyse more deeply the differences in genetic programs between the two species.

R: Thanks for the comments and suggestions. To improve the integration of the manuscript, we focused on three main findings (temporal hierarchy, spatial dynamics and allele-specific expression of ZGA in *Ciona*), which are logically connected and were revealed using the unique *Ciona* hybrid system. We have reorganized the manuscript. For example, we deleted the gene age part and simplified the gene regulation part in the revision.

In addition, to provide more solid evidences for each aspect of our main findings, we performed additional experiments. For example, we added the CUT&Tag data of H3K27ac and Pol II to support the two waves of ZGA, carried out single cell RNA-Seq of reverse cross hybrid embryos and cell fate transfer experiments to support the new model for ZGA, and conducted motif flip experiments to strengthen the mechanisms of ASE. We hope these additional results will help to integrate the manuscript and strengthen the deeper understanding of our findings.

Regarding to the differences in genetic programs between the two species, we agreed with the reviewer's suggestion. Our results revealed that the species-biased allele specific expression was related to the genetic differences between the two species. We have added the comparative analysis of the activity of cis-regulatory motifs between *C. robusta* and *C. savignyi* by motif flip experiments in the revised manuscript. Nevertheless, regarding to temporal-spatial dynamics of ZGA, the forward and reverse cross hybrids showed similar patterns, indicating that the zygotic genome activation process are largely conserved between the two species. The conservation makes it suitable for using both directional hybrids to study the zygotic genome activation. In the future, study on the genetic difference between the two species using hybrid samples will be an interesting direction.

Some parts are difficult to follow because of vague or insufficient explanations. This applies in particular to the analysis of the regulatory relationships between maternal and successive waves of zygotic genes (lines 171 to 187). After reading the main text and the materials and methods, I could still not really understand what had been done and whether it made any sense. I was not even sure that the « motif scanning » section was relevant as the 244 genes described here do not seem to match those of lines 171 and following. Providing a list of the genes being analysed would be a first step. I was also surprised that the authors searched for motifs in the 100-300 bases immediately upstream of what I understood to be the initiator methionine. Many *Ciona* genes are trans-spliced with 5' outrons. The TSS may thus be located some distance away from the initiator methionine (see Matsumoto et al., Genome research, 2010). It would be much preferable to scan open chromatin regions within and upstream of the genes of interest (see Madgwick et al., Dev. Biol. 2019).

R: The TF genes and their motif bearing genes analyzed in this part were used for regulatory relationship analysis. For example, 71 maternal TFs and 301 motif bearing

genes can be identified based on JASPAR database. The methods were included in “Regulatory relationship analysis” part in Methods. The TFs in minor waves and their motif bearing genes in major wave have been listed in supplementary Table 4 in the revision. 244 genes were not used for regulatory relationship analysis, but were screened as species-biased genes, which were included in supplementary Table 15. The methods are included in “Motif scanning” part in Methods. Sorry for the unclear statement in the part of regulatory relationships, we have rewritten this part in the revised manuscript (from line 182 to line 191).

We agreed with the reviewer that TSS may be located some distance away from the initiator methionine. Therefore, we first conducted upstream sequence (0-1000 bp upstream of initiator methionine) identity analysis on gene homolog pairs. The result revealed significant sequence differences for 100-300 bp sequence window (Extended Data Fig. S22). Therefore, we conducted motif scanning analysis targeting at 100-300bp region and selected binding sites that may lead to expressional bias.

Thank you for the indication of trans-splicing events in *Ciona*. It was possible that the TSS were located some distance away from the initiator methionine trans-spliced genes. We checked the spliced leader sequences ¹ in our bulk RNA-seq data and found that there was very low ratio (0.001%) of reads containing the spliced leader sequence of *C. robusta*. In addition, it seems that most zygotic mRNA is not trans-spliced in tunicate species according to previous studies in *Oikopleura dioica* ^{2,3}. Furthermore, as the reviewer suggested, we provided the CUT&Tag scanning of both Pol II and H3K27Ac, which do help to provide the position of regulatory region of our interested genes (Fig 4c).

The results are frequently stated but not interpreted. It is therefore difficult to tell apart mere confirmatory results from true interesting novelties. For instance, do the authors consider the discovery that different classes of genes have different « ages » according to phylostratigraphy as a major result or as a confirmation in *Ciona* of what has been found in other systems?

R: The trend of gene age during ZGA indeed has been discovered in other systems ⁴. Here, we showed the similar trend in urochordate using hybrid system. In order to keep the integration of the manuscript, we have deleted this part in the revised manuscript.

Are the genes composing the modules reported line 150 and following orthologous between species? or do the modules consist in different genes in the two species?

R: Because the samples were from hybrid systems, the genes in each module include genes from both species. These genes are composed of both orthologous and different genes between the two species. For Cr ♀ × Cs ♂, there are 172 genes from *C. robusta* and 111 genes from *C. savignyi* in module 1 (minor wave), and 98 of 172 and 48 of 111 have orthologous genes between species. There are 1,054 genes from *C. robusta* and 3,301 genes from *C. savignyi* in module 2 and 3 (major wave), and 543 of 1,054 and 2,151 of 3,301 have orthologous genes between species. For Cs ♀ × Cr ♂, there are 110 genes from *C. robusta* and 292 genes from *C. savignyi* in module 1 and 39 of 110 and 166 of 292 have orthologous genes between species. There are 1,833 genes from *C. robusta* and 1,264 genes from *C. savignyi* in module 2 and 3, and 1,129 of 1,833 and 696 of 1,264 have orthologous genes between species. These data have been added in supplementary Table 2.

Lines 232-23 : I am not convinced of the claimed lack of correlation between the timing of activation of housekeeping genes and the time elapsed since the last cell division. For instance at the 64-cell stage these genes are preferentially expressed in endoderm and notochord (Fig. 3g) and the precursor cells in these two tissues are the ones that have been around for the longest time (since the 44-cell stage).

R: Here we mean they are not fully supported by the time elapsed since the last cell division. The correlations are indeed suitable for endoderm and notochord cells at the 64-cell stage as the reviewer mentioned, but not for all the cell types such as mesenchyme cells. We have toned down in line 245.

Minor comments:

Important concepts should be better defined. For instance what is MBT precisely? If it is defined by the desynchronisation of cell divisions, then this occurs in *Ciona* just after the 16-cell stage (see Nishida's early work). Also the 2 « waves » of zygotic transcription that define ZGA should be better defined, and called the same way throughout the ms (currently they are referred to as first and second waves, and as minor and major waves). Also how do the authors define « housekeeping » genes?

R: MBT (mid-blastula transition) were indicated by asynchronous cell divisions and longer cell cycles, and followed by gastrulation. We agreed that the asynchronous cell divisions occurred just after 16-cell stage in *Ciona*, however, the asynchronous division in *Ciona* was not followed by gastrulation, which indicated a not precise MBT in *Ciona*.

For the two waves of ZGA, we should keep the same way throughout the manuscript. We have changed “first wave” and “second wave” into “minor wave” and “major wave” to keep the consistence in the manuscript.

The housekeeping genes were commonly described as essential for cellular existence, and stably expressed irrespective of developmental stage. The housekeeping genes in this analysis were screened by the standard that genes are maternally expressed and have been reactivated at late-neurula stage. We have added the description in Line 204 - 208.

Line 92 : *Ciona savignyi* is not « distributed across most coasts around the world » : WORMS indicates that it is mostly found in the North Pacific ocean.

R: The reviewer is right. We have revised this sentence into “Both *C. robusta* and *C. savignyi* are tunicate species with similar morphologies, and are broadly distributed across North Pacific Ocean” in line 101.

While *Halocynthia roretzi* counts 110-cells at the onset of gastrulation, *Ciona* and other phlebobranch species count 112-cells at the same stage (due to the precocious division of the A7.6 cell pair ; see Tassy et al. Curr. Biol. 2006 ; Guignard et al. 2020).

R: Thank you for pointing this out. The reviewer is right. We have clarified it in the manuscript “110-cell stage (112 cells at this stage)” in line 415.

“110-cell” stage stands for the initial gastrula stage in *Ciona*. Here we used “110-cell” stage to indicate the cleavage stage in *Ciona*, which is also used in other publications^{5,6}, not the exact cell number.

Lines 74 and 197: The idea that ascidians develop using a mosaic strategy (i.e. one in which cells do not communicate with each other) has been shown wrong by many authors (starting with H. Nishida). The authors are right that ascidian embryos develop with invariant lineages (Conklin, Satoh, Nishida, Christiaen, Lemaire labs among many others), but this is different from « mosaic » development. The term mosaic should be avoided in this context.

R: We agreed with the reviewer that cells during early development of ascidians are not totally independent. Therefore, we avoided using “mosaic” term in the context in revision.

Lines 248: Why do the authors describe Nodal and Ephrin genes as « stiffness genes » ?
Is there a reference for this?

R: According to the previous reports, *Nodal* and *Ephrin* are related to cell stiffness^{7,8}. They demonstrated the role for Nodal-controlled cell-cortex tension in germ-layer organization and downregulation of *Eph* results in the decreased cell stiffness. We have added the references in line 254.

Lines 289 and Figure 4d : I do not understand why the authors only show binding sites for Fox family genes ?

R: Here We analyzed 86 publicly available motif profiles for motif scanning from JASPAR database, not only for Fox family genes. Since we found the significant sequence differences for upstream 100-300 bp among the 1000 bp (Extended Data Fig. S22). Therefore, we conducted motif scanning analysis targeting at 100-300 bp region and screened the different binding sites. There are indeed many different binding sites from Fox family, but also other binding sites such as *otp* and *Sox* family. The binding sites of allelic expression genes were listed in supplementary Table 15 and the diagram for motif distribution was shown in Extended Data Fig. S23.

Lines 226-227: I do not understand this sentence.

R: Here we mean housekeeping genes were not expressed simultaneously in each cell during early development. This suggested that ZGA in *Ciona* are not correlated to exact time and did not follow the developmental clock model. We have revised this sentence as “Overall, these results indicated that zygotic housekeeping genes were asynchronized reactivation in hybrid embryos, and thus did not follow the maternal clock model.” in line 238 - 240.

Lines 263-264: I do not understand this sentence.

R: Here we mean the maternal deposited TFs had advantage at recognizing enhancers from maternal DNA. We have revised this sentence in line 281 - 282.

Line 363: what does « alone » mean ?

R: Sorry for the mistake. It should be “along” here.

Reviewer #2 (Remarks to the Author):

SUMMARY

In this study, Wei et al. used forced hybrids between two ascidian species, *Ciona robusta* and *Ciona savignyi*, which have widely divergent genomes, to profile the transcriptomes of embryos at successive early stages, and identify parent specific gene expression, to gain insight into the genome-wide features of zygotic gene activation (ZGA) in ascidians. Focusing on paternal vs. maternal gene expression of housekeeping genes, the authors propose that ZGA occurs in two waves. Using single cell RNA-seq, and chromogenic in situ hybridization, the authors uncover lineage and spatial differences in the relative proportion of paternal-genome-derived housekeeping gene expression, which they relate to asynchronous ZGA. The authors propose that neither the nucleocytoplasmic ratio nor the timing after cell division account for these spatiotemporal differences, but cell stiffness. Finally, the authors reveal species-specific differences in allelic gene expression presumably attributable to differences in cis-regulatory sequences.

KEY MERITS AND GENERAL EVALUATION

Key merits of the study include the generally important problem of ZGA in tunicates, which remains understudied, as is the notion of asynchronous and lineage-specific zygotic genome activation. The authors decidedly leverage a unique and powerful biological feature, namely the ability to generate viable first-generation hybrids between species with very divergent genomes but very conserved early embryogenesis. The authors also decisively used modern transcriptomics approaches, including from single cells. The datasets generated are of potential high value, and the notion that ZGA occurs asynchronously in the various lineages of early embryos is well supported by the single cell RNA-seq and in situ data. Nevertheless, there are a many marked shortcomings throughout the study, which strongly reduced enthusiasm for this manuscript.

CONCERNS

- One of the key conclusions of the paper, that there are two waves of ZGA, is not convincingly supported by the data and analysis. That conclusion appears rather abruptly at the end of page 6, relying on plots shown in Ext Data Fig S2, which show

complex dendrograms without much explanation. Moreover, the heatmaps shown on Fig. 2d, e could as well support the notion that ZGA is quite continuous, especially considering that this is bulk RNA-seq and the following section emphasize lineage specific timing.

R: We agreed that ZGA process is indeed continuous. Nevertheless, the statistics on paternal gene ratio (Fig. 2b) and *de novo* transcribed gene number (Fig. 2c) showed the varying growth rates, which indicated the two ZGA waves (minor and major wave) during *Ciona* early embryogenesis. In addition, we defined two ZGA waves indicated the initiation of zygotic gene expression and housekeeping gene reactivation, respectively in this manuscript. To clarify our statement, we have rewritten this section from line 122 to line 153.

- Fig. 2h is not supported at all and should be removed. There is no analysis of maternal gene product degradation in the paper.

R: Thanks for the suggestion. We have deleted it in the revised manuscript.

- The metrics used to evaluate zygotic genome activation, namely the proportion of paternal RNA for each gene is flawed in several ways: it assumes, which may be largely true, that the sperm does not contribute to adding paternal RNA to the zygote. Also, because it is relative the total amount of RNA for each gene, including maternal RNA that may be degraded, the metric convolves both RNA synthesis, for each allele, and RNA degradation, from each genotype. This reviewer appreciates the attempt to get at genome activity with CUT&Tag for RNAPII and histone marks, but this is not primarily used. An unexploited option is to leverage intronic reads to more accurately measure RNA synthesis from each parental genome.

R: Thanks for the comprehensive consideration. As the reviewer suggested, we have added the CUT&Tag experiments of both H3K27ac and RNA Pol II at different stages (including 16-, 32-, 64-, and 110-cell and early neurula stages) to provide the cis-regulatory landscape of both species. The CUT&Tag results from hybrid ascidians supported the results from RNA-seq data (Fig. 2e, Extended Data Fig. S5).

In addition, we analyzed the intronic reads as reviewer suggested. Bulk RNA-seq data of both hybrid directions were aligned against their parental species genome by Hisat2. The feature type was set to 'intron' and intronic reads ratio were calculated by successfully assigned alignments/total alignments from results of featureCounts. The

results showed a similar trend with the statistics of proportion of paternal RNA (Extended Data Fig. S3).

- The attempt to explain spatial/lineage differences in proportion of paternal transcripts is very tentative. Specifically, Ext Data Fig 16 does not mention ZGA at all, and there is no attempt to perturb either of the parameters (N/C ratio, timing after division, stiffness) and assay ZGA to try and establish a causal relationship. The conclusion that stiffness of endoderm progenitor cells somehow promote activation of the zygotic genome is thus not supported by the evidence presented.

R: Ext Data Fig 16 was used to show that ZGA in hybrid *Ciona* was not fully correlated with cell volume or the time elapsed since the last cell division. We would like to clarify that, here we mean that cell fate might be the reason for asynchronized ZGA. The cell stiffness could be a consequence of cell fate, rather than the cause of ZGA. We have re-organized the statements of this paragraph.

The reviewer raised an important issue, unfortunately it is very difficult in *Ciona* to perturb these parameters directly and then to assay ZGA. We have tried our best to conduct cell stiffness measurement experiment after perturbing cell fates in *Ciona* to further address the possible mechanisms of the dynamics of ZGA in the manuscript. The result showed a decreased cell stiffness when notochord cell fates were transferred, which referred a correlation between cell fates and cell stiffness. Combining with the correlation between ZGA and cell fates from our previous analysis, we inferred a new possible model for ZGA in *Ciona* embryogenesis. The related results were showed in Fig. 3h and line 257 to line 267.

- The last part connecting allele-specific zygotic gene expression to differences in cis-regulatory sequences is potentially interesting, but suffers from several caveats as well:
 - o the point is almost trivial and supported by studies in other systems; including the chosen cis-regulatory DNAs that have been studied in *Ciona* and show different results.
 - o the in silico cis-regulatory analysis is not convincing as it focuses on very short upstream regions, ignoring introns, more distal elements and generally the accessibility data available for *Ciona*; the motifs used are tentative and it is not clear to this reviewer how the authors can effectively distinguish between all the Fox motifs (Ext Data Fig 20); finally the conclusion is not tested by changing candidate binding sites in cross-species reporter assays.

R: We supplied the CUT&Tag experiments of both H3K27ac and RNA Pol II at different stages (including 16-, 32-, 64-, and 110-cell and early neurula stages) to provide the cis-regulatory landscape of both species. The modification peaks from hybrid ascidians showed consistent trends with RNA-Seq results (Extended Data Fig. S21), which also support more species-biased allele-specific activation of gene expression.

In this section, we first conducted upstream sequence (0-1000 bp upstream of initiator methionine) identity analysis on gene homolog pairs. The result revealed significant sequence differences for 100-300 bp sequence window (Extended Data Fig. S22). Therefore, we conducted motif scanning analysis targeting at 100-300 bp region and selected binding sites that may lead to expressional bias. The identification of Fox motifs was based on the data from JASPAR database, which provide the motifs for transcription factors in *Ciona*.

As the reviewer suggested, in order to further validate the relationship between species-biased expression and cis-regulatory elements, we compared the activity of regulatory motif from orthologous genes between *C. robusta* and *C. savignyi* by motif sequence flip experiment. The results of *titin* and *FoxF* genes all showed expression in adding cis-regulatory elements from *C. robusta* into promoter region of *C. savignyi*. This indicated the existence of binding-motif would influence the allele specific expression in hybrid ascidians, which also support our previous finding that ASE effect is attributed to the divergence of cis-regulatory elements of two ascidian species. The results were shown in Fig 4d and Extended Data Fig. S23-24.

- Ultimately, the paper gives the impression that it attempts to include too many points without strong or informative support for the main conclusions. Specifically, the section on the presumed evolutionary ages of genes is potentially interesting but under-developed and not validated; Est Data Figs. 8-10 are not particularly informative. The study would strongly benefit from focusing on the key points and strengthening the experimental and analytical support.

R: Thank you for the comments. To improve the integration of the manuscript, we focused on three main findings (temporal hierarchy, spatial dynamics and allele-specific expression of ZGA in *Ciona*), which are logically connected and supported each other in the revision, and re-organized the manuscript. For example, we deleted the gene age part and simplified the gene regulation part in the revision. Data Figs. 8-10 were listed

to show the conserved regulatory relationships between the two species, since we have rewritten this section, we also deleted them in revised manuscript to keep the integration of the manuscript.

In addition, to provide more solid evidences for each aspect of our findings, we performed additional experiments. For example, we added the CUT&Tag of H3K27ac and Pol II data to support the two waves of ZGA, carried out single cell RNA-seq of reverse hybrid embryos and cell fate transfer experiments to support the new model for ZGA, and conducted motif flip experiments to strengthen the mechanisms of ASE. We hope these additional results will help to integrate the manuscript and strengthen the deeper understanding of our findings.

Reviewer #3 (Remarks to the Author):

In this manuscript by J. Wei, W. Zhang, L. Leng, A. Jiang and colleagues, the authors are undertaking on a long-standing question regarding the critical stage of ZGA during metazoan embryogenesis, using an elegant hybrid model of ascidian species. The authors were able to demonstrate the minor and major ZGA waves in early development of the hybrid embryos using mostly gene expression analysis approaches. Moreover, the authors show that, in ascidian development, the spatial gene expression heterogeneity during ZGA is correlated to cell fate and cell stiffness rather than previously offered models. The authors also provide some evidence for regulatory connection between maternal and zygotic genes as well as ASE in a species, rather than parental, -related manner. The authors' findings linking ZGA timing to cell fate and stiffness, as well as, the discovery of species-biased allelic gene activation during ZGA, are of high interest to the field.

Although the gene expression analysis approaches in this manuscript hold a potential to unravel interesting evolutionary insights into ZGA regulatory mechanisms, overall, the manuscript lacks a deeper analysis into the claimed mechanisms as well as experimental evidences to support some of the statements and interpretations that are made in the text body. As the vast majority of the claims in the presented manuscript are based on RNA-seq data, without more detailed analysis and further experimental follow-up, I suggest that the authors should rephrase and tone down some of them, and widely edit the figures to make them easier to follow.

R: Thanks for the comments and suggestions. To improve the integration of the manuscript, we focused on three main findings (temporal hierarchy, spatial dynamics and allele-specific expression of ZGA in *Ciona*), which are logically connected and supported each other in the revision and re-organized the manuscript. For example, we deleted the gene age part and simplified the gene regulation part in the revision. The figures also have been revised.

In addition, to provide more solid evidences for each aspect of our findings, we performed additional experiments. For example, we added the CUT&Tag of H3K27ac and Pol II data to support the two waves of ZGA, carried out single cell RNA-seq of reverse hybrid embryos and cell fate transfer experiment to support the new model for ZGA, and conducted motif flip experiment to strengthen the mechanisms of ASE. We

hope these additional results will help to integrate the manuscript and strengthen the deeper understanding of our findings.

While hybrid *Ciona* is presented as a very intriguing model to study ZGA and ASE by being the most distant hybrid embryos to date, it is only used as a descriptive system at this stage. It is unclear and necessary to explain with more data to the general reader how the early embryogenesis is proven to be intact in the hybrid *Ciona* given that the defects of the embryos start very early in organogenesis. A comprehensive and comparative characterization of early embryogenesis is needed between hybrid and non-hybrid embryos to demonstrate that the ZGA in hybrid embryos are intact and physiologically relevant. Furthermore, at least a proof-of-concept experiment that targets specific gene expression to intervene ZGA dynamics, cell fate or stiffness is necessary to confirm the causality in the interpretation of the results, apply the hybrid chordate system more widely and claim it as a novel system to study ZGA dynamics.

R: Thank you for the suggestion. In the revised manuscript, we have added the statistics data of developmental failure rates during early development of self-cross and hybrid *Ciona*. The results were added in line 108 to line 117 (Supplementary Table 1, Extended Data Fig. S2). There was no visible difference in the embryogenesis process between forward cross and reverse cross embryos before the tailbud stage. We think the conservation makes it suitable for using both direction hybrids to study the zygotic genome activation.

The reviewer raised an important issue, unfortunately it is very difficult in *Ciona* to perturb these parameters directly. Nevertheless, in order to prove the causality between cell fate and stiffness, we conducted the experiments of cell stiffness measurements after perturbing cell fates in *Ciona*. The result showed a decreased cell stiffness when notochord cells were transferred (Fig 3h), which referred a correlation between cell fates and cell stiffness. Combining with the correlation between ZGA and cell fates from our previous analysis, we inferred a new possible model for ZGA process.

To further address some of the concerns, more substantial analyses are needed, such as detailed comparison of genes in different stages or waves, use of scRNA-seq data from both $Cs_{\text{♀}} \times Cr_{\text{♂}}$ and $Cr_{\text{♀}} \times Cs_{\text{♂}}$ hybrid crosses, validation of species-biased allelic gene activation results with additional methods such as H3K27ac and RNA PolIII ChIP-seq, and demonstration of the species-dominant TFs with CUT&RUN data. Overall,

the authors' work would benefit from a more in-depth analysis to strengthen their findings and interpretations.

R: Thanks for the suggestion, we have conducted the single-cell sequence of the reverse cross $Cs_{\text{♀}} \times Cr_{\text{♂}}$ hybrids at 64-cell, 110-cell and early neurula stages, we tried our best but only 110-cell stage sample was successfully accomplished (Extended Data Fig. S13; Supplementary Table 10). The paternal gene ratio analysis of reverse cross embryos at 110-cell stage also indicated asynchronous ZGA process in different cell types (Extended Data Figs. S14), which is consistent with our previous conclusion. The endoderm and notochord cells also exceeded than other cell types in paternal gene ratio (Extended Data Figs. S15; Supplementary Table 11).

We also supplied the CUT&Tag experiments of both H3K27ac and RNA Pol II at different stages (including 16-, 32-, 64-, and 110-cell and early neurula stages) to provide the cis-regulatory landscape of both species. The results support more species-biased allele-specific activation of gene expression (Extended Data Fig. S21).

We agreed that CUT&RUN of transcription factors would be benefit to verify their allele-specific manner. However, it is regretful that we cannot get suitable antibodies for the CUT&RUN experiment of the TFs. Hopefully these experiments could be done in the future.

Major points:

1. Line 99-100 - What does "slightly higher development failure rate" mean? In Figure 1c, tail bud stage looks quite different between reciprocal crosses. Could the authors provide more evidence than morphology by brightfield microscopy to claim "no significant difference in embryogenesis". Subsequently, can the authors provide quantification and/or description of the developmental failure? Could the authors provide more pictures as Extended Data? The statements are difficult to judge based on a single picture at each stage without replicates and more details on *Ciona* embryogenesis.

R: Thanks for the suggestion. We have added the statistics data of developmental failure rates during early development of self-cross and hybrid *Ciona*. The developmental observations and statistics results were added in line 108 – 117 (Supplementary Table 1, Extended Data Fig. S2). Although one direction of the hybrid embryos ($Cs_{\text{♀}} \times Cr_{\text{♂}}$) had a higher developmental failure rate compared to the cross ($Cr_{\text{♀}} \times Cs_{\text{♂}}$) at early neurula stage (Supplementary Table 1, Extended Data Fig. S2). There was no visible

difference in the embryogenesis process between forward and reverse cross embryos before the tailbud stage.

2. Line 101 - 'which is distant from our research window' - The research window should be introduced for ZGA in Ciona, therefore, either letting the reader to judge for themselves or guide the reader whether it is distant enough.

R: Here the research window means the early development stages which ZGA occurred. It referred to the stage from 1-cell to gastrula stage in our manuscript. we have revised it in line 116.

3. Figure 1d is not mentioned in the text.

R: Figure 1d indicated our procedures for gene expression profiles by bulk RNA-seq and single-cell RNA-seq. We have cited it in the revised manuscript in line 119.

4. Figure 2a,b,c are confusing and hard to follow. Does each color represent each reciprocal hybrid or each genome? The figure legend also does not give enough explanation. The red arrows seem to depict each ZGA stage but this should be written in the Figure legend.

R: Thanks for the suggestion, we have changed the color code for self-cross and hybrid animals. And we also added the descriptions in figure legends. We hope the revised colors could make it more distinct to show the hybrids and each species genome. We also added the interpretation in the revised figure legends. The red arrows indicated the minor and major ZGA which corresponded with the initiation of zygotic gene expression and housekeeping gene reactivation, respectively.

5. Line 114 - How is the paternal ratio for both crosses at stage 16-cell is 0.02% when the plot in Figure 2a appears to have much more transcripts in one cross compared to the other? Overall, it seems that the Cr♂ expression is always higher, given the 'higher developmental failure rate' mentioned in Figure 1, I think that this should be further explored.

R: The paternal ratio was calculated according to the reads mapped to the genes while the Figure 2a was calculated according to the reads mapped to the whole genome. We have clarified this in the revised manuscript.

There are 18,191 genes annotated on the reference genome of *C. robusta*⁹ while 12,172 genes were annotated on the reference genome of *C. savignyi*¹⁰. When we calculate the number of paternal genes, there are more expressed genes in $Cs_{\text{♀}} \times Cr_{\text{♂}}$ hybrid than in $Cr_{\text{♀}} \times Cs_{\text{♂}}$ hybrid with regard to same developmental stage. However, we think the discrepancy probably not due to thresholding or any difference in pooled sample size or depth of the RNA-seq. The differences on assembly and annotation quality of the two genomes could be the reason. Hence in this manuscript, we didn't pay much attention on the genetic differences between the two species. We have changed the gene number into the ratio of gene number in order to avoid misleading (Fig 2c).

6. Line 121 - Is it possible for the maternal genome to have a stable expression bias over the paternal genome in any of these species or hybrids? Is it safe to interpret that "maternally inherited transcripts might still be retained until the neurula stage" without having data in the further stages?

R: Yes, we agreed with the reviewer. Since we didn't check the retaining of maternally inherited transcripts, we cannot conclude that "maternally inherited transcripts might still be retained until the neurula stage". We have removed the statement in the revised manuscript.

7. In Figure 2c, blue embryos show higher de novo gene numbers for most timepoints. Does this mean that, in total, do they express more number of genes? If so, are these additionally scored genes lowly expressed genes? If so is the discrepancy due to thresholding or any difference in pooled sample size or depth of the RNA-seq?

R: When we calculate the number of paternal genes, there are indeed more genes in $Cs_{\text{♀}} \times Cr_{\text{♂}}$ hybrid than in $Cr_{\text{♀}} \times Cs_{\text{♂}}$ hybrid with regard to same developmental stage. However, we don't think these situations are related to additionally scored genes lowly expressed genes, and the discrepancy probably not due to thresholding or any difference in pooled sample size or depth of the RNA-seq. We used the same threshold and similar sequence depth.

There are 18,191 genes annotated on the reference genome of *C. robusta*⁹ while 12,172 genes were annotated on the reference genome of *C. savignyi*¹⁰. The differences on assembly and annotation quality of the two genomes could be the reason. We have

changed the gene number into the ratio of gene number in order to avoid misleading (Fig 2c).

8. Line 137 - 'Combining the appearance of gene and expression characteristics' - what does it mean? Some experimental details will help the reader.

R: Here “the appearance of gene” we mean the de novo transcribed genes. And the “expression characteristics” we mean the gene expression characteristics of each sample. Sorry for the misleading, and we have deleted this sentence in the revised manuscript.

9. The aim and presentation of results in supplemental figure 3 are unclear. Please provide clustering of the CUT&Tag peaks (to show what those few genes that are indeed bound are). Is PolIII more enriched in the gene body compared to TSS? The tracks provided are without scales in panel b, making it impossible to assess the enrichment of the peaks. How many replicates were used? Why was only one hybrid and one stage tested?

R: Thank you for the suggestion. Here we aimed to ensure the gene expression characteristics from RNA-seq data and whether maternal orthologs were also transcribed as paternal orthologs. We have supplied the CUT&Tag experiments of both H3K27ac and RNA Pol II at different stages (including 16-, 32-, 64-, and 110-cell and early neurula stage) to provide the cis-regulatory landscape of both species. The results from hybrid ascidians supported the RNA-seq results (Fig. 2e, Extended Data Fig. S5), and also support the species-biased allele-specific activation of gene expression (Extended Data Fig. S21).

We have replaced the previous CUT&Tag data with new datasets from five developmental stages and at least two replicates (Metadata of datasets can be browsed in NCBI with accession number of PRJNA1002800). The Pol II enrichment analysis (Extended Data Fig. S21) showed that the Pol II enrichment was centered by TSS.

10. Line 156 - "regulation" should be detailed in the text. What is it referring to?

R: The “regulation” here referred to the GO term of biological process including regulation of transcription, regulation of nucleic acid, regulation of RNA biosynthetic process. We have added the detailed description in the revised manuscript in line 174 - 178.

11. Line 158 - In Figure 2d-e, what are the genes in each module and how much overlap are there for each module between reciprocal hybrids?

R: The genes in each module were clustered according to their expression using WGNCA analysis. The genes in each module were listed in supplementary Table3.

There are 402 genes for $Cs_{\text{♀}} \times Cr_{\text{♂}}$ and 283 genes for $Cr_{\text{♀}} \times Cs_{\text{♂}}$ in module 1 (minor wave genes), respectively. 71 genes are overlapped between reciprocal hybrids in module 1. There are 3,097 genes for $Cs_{\text{♀}} \times Cr_{\text{♂}}$ and 4,355 genes for $Cr_{\text{♀}} \times Cs_{\text{♂}}$ in module 2 and 3 (major wave genes), respectively. 627 genes are overlapped between reciprocal hybrids in module 2 and 3. We have added the results in supplementary Table 3.

12. It appears that Fig 2g is not mentioned in the text at all. Additionally, in Fig 2g, the definitions for minor and major waves are confusing as they refer to the first and second wave, respectively. Moreover, the genes from the first and second wave are only defined in Fig 2 and not used in the following analysis. Can the authors clarify if genes from first and second wave show consistent results regarding the relation to cell stiffness and species-biased ASE?

R: Thanks for the comments. Fig 2g has been removed in the revised manuscript. We should keep the same way throughout the manuscript. We have changed first wave and second wave into minor wave and major wave, respectively to keep the consistence.

The stiffness data was collected from previous research¹¹. Unfortunately, we can only find data at 76-cell stage. There are no stiffness data for 64-cell stage. We think that data at 76-cell stage could also be representative.

The species-biased genes for different stages were shown in Extended Data Fig. S19. Our data indicated that the ASE are not consistent in early stages and late stages.

13. In Figure 3, it is unclear which hybrid cross is used for scRNA-seq, $Cs_{\text{♀}} \times Cr_{\text{♂}}$ or $Cr_{\text{♀}} \times Cs_{\text{♂}}$? The scRNA-seq of the both hybrid crosses is needed to validate all the results in Fig 3. While the correlation of cell stiffness to ZGA timing is an interesting topic, the analysis presented is relatively shallow. For example, it may be more appropriate to use a scatterplot with R-value and P-value to show the correlation rather than a boxplot (Fig. 3h). Additionally, what happened to the 64 single cell data, and do they show similar results?

R: In Figure 3, $Cr♀ \times Cs♂$ hybrids were used for scRNA-seq. The scRNA-seq of $Cs♀ \times Cr♂$ hybrid crosses at 110-cell stage was supplemented to validate the results (Extended Data Fig. S13; Supplementary Table 10). The paternal gene ratio analysis also indicated asynchronous ZGA process in different cell types (Extended Data Figs. S14), which is consistent with our previous conclusion. The endoderm and notochord cells also exceeded than other cell types in paternal gene ratio and indicated similar results with $Cs♀ \times Cr♂$ hybrid crosses (Extended Data Figs. S15; Supplementary Table 11).

The stiffness data was collected from previous research¹¹. Unfortunately, we can only find data at 76-cell stage. There are no stiffness data for 64-cell stage. We cannot acquire the corresponding results of both paternal gene ratio and cell stiffness for one cell to show the correlation by scatterplot.

14. Figure 4a is difficult to read, and a boxplot may better show the pattern of the four groups. It is not clear whether the 341 pairs used in Fig 4a are identified by including all stages or only late stages (second wave?). Furthermore, a more comprehensive analysis of species-biased ASE should be done for different stages or for first and second wave genes. Are species-biased ASEs consistent in different stages or stage-specific?

R: Thanks for the suggestion, we have added the boxplot in Extended Data Fig. S19. The data in Fig. 4a showed the allelic specific expression from the 110-cell to late-neurula stage (major wave). The species-biased ASE for different stages were shown in Extended Data Fig. S19. Our data indicated that the ASE are not consistent in early stages and late stages. The minor wave genes (Extended Data Fig. S19a) tended to be *C. robusta*- or maternal-dominant, which might be attributed to that maternal deposited TFs had advantage at recognizing enhancers from maternal DNA. Only the expression of major wave genes tends to species-biased.

15. In Figure 4d, It is unclear whether it is plotted for a specific gene or is just a schematic diagram. The authors claim that "the distribution of the trans-regulating element binding motifs of FoxF and titin in homolog genes in *Ciona* showed a *C. robusta*-dominant pattern, whereas motifs regulating IAP2 and ZicL showed a *C. savignyi*-dominant pattern." However, it is unclear whether any motif enrichment analysis was performed, and it is challenging to claim "dominant" without performing

any ratio calculation or statistical tests. A more systematic analysis is needed to make such strong claims, such as performing CUT&Tag of FoxF, titin, IAP2, and ZicL to verify whether they truly bind to those genes in an allele-specific manner.

R: Figure 4d was drawn according to the motif positions. It is a schematic diagram for species-biased genes. The pattern was defined according to the motif number identified in the two ortholog genes, so we didn't perform the enrichment analysis. We have revised this section in line 307 – 315.

Thanks for the insightful suggestion. We added the CUT&Tag analysis of H3K27ac and Pol II to validate their binding sites. The H3K27ac and Pol II occupancy detected by CUT&Tag at early neurula stage of titin in reverse cross embryos also showed that the signal aligned in 0-1000 bp upstream in *C. robusta* exceeded that in *C. savignyi* distinctly after normalization (Fig. 4c).

We agreed that CUT&Tag data of upstream TFs of *FoxF*, *titin*, *IAP2*, and *ZicL* was also useful to verify their allele-specific manner. However, it is regretful that we cannot get suitable antibodies for the TFs. Hopefully we could perform the experiment in the future.

16. The authors could also use H3K27ac and RNA PolIII data to validate the species-biased ASE findings.

R: We have added the CUT&Tag experiments of both H3K27ac and RNA Pol II at different stages (including 16-, 32-, 64-, and 110-cell and early neurula stages) to provide the cis-regulatory landscape of both species. The results from hybrid ascidians also support the minor and major expression waves, and also support more species-biased allele-specific activation of gene expression.

In $Cs_{\text{♀}} \times Cr_{\text{♂}}$ embryos, Cr-dominant gene group showed consistence dominance by Cr allele over Cs allele. While in Cs-dominant gene group, with the proceeding of development and activation of genes, Cr allele profile were overtaken by Cs allele profile at 110-cell and early neurula stages. Similarly, the Maternal-biased gene group showed similar trend with Cs-dominated gene group, which is consistent with the fact that *C. savignyi* as female parent (Extended Data Fig. S21).

17. Line 550 - In the methods, what is the 'Statistics analysis' referring to? It seems that the manuscript contains much more elaborate statistics.

R: The statistics analysis referred to the significance test used for the statistics analysis of paternal transcript ratio in each cell type. Sorry for the confusion in this part. We have deleted this paragraph and revised the description in line 491 – 492.

Minor points:

1. The authors should use either "massive" ZGA or "major" ZGA throughout the manuscript for consistency.

R: Thanks for the suggestion. We should keep the same way throughout the manuscript. We have changed first wave and second wave into minor wave and major wave in order to keep the consistence.

2. Line 128 - All genes should be presented by their full name, when introduced for the first time in the text.

R: We have added the full name “60S ribosomal protein L11, 60S ribosomal protein L13, transport protein SEC61, KRR1 small subunit processsome component, heterogeneous nuclear ribonucleoprotein D0” for gene *60SL11*, *RPL13*, *SEC*, *KRR* and *HNRN* in Line 233 – 236.

3. Line 154 - It should be "gene" instead of "gen"

R: Sorry for the mistake, we have revised this word in line 174.

4. Line 169 - "The specificity of ZGA might be related to the hourglass model of developmental evolution" - this is not a result. I will suggest leaving it to the discussion.

R: Thanks for the suggestion. In order to keep the integration of the manuscript, we have already removed the gene age section in the revised version.

5. It would be helpful if the authors could label the hybrid cross and developmental stage in all figures (i.e., Fig3, Fig 4d,e, Extended Fig 3b).

R: As the reviewer suggested, we labeled the hybrid cross and developmental stages in the figures. We have also revised the figure legends.

6. Line 210 - it should be "in each cell type" instead of "in each cell".

R: Thanks for pointing this out. We have revised this phrase in line 218.

7. Line 220, it should be "validate" instead of "valid."

R: Sorry for the mistake, we have revised this word in line 231.

8. Line 227 - 'the developmental clock model' - This is the second time that this appears. The authors should address this in the introduction.

R: Thanks for the suggestion, we have added the content about developmental clock model in Introduction in line 58 - 60 "In addition to these models, the maternal-clock model presumes that fertilization initiates a biochemical cascade that serves as a molecular timer".

9. Lines 235-238 - a very long and unclear sentence. Please rephrase.

R: Thanks for the suggestion, we have rephrased the sentence into "In addition to the variation in cell size, a new explanation has been proposed for the different timing of ZGA in *Xenopus* embryos. This explanation concerns the expression timing of specific germ-layer genes, indicating that the animal pole-derived ectoderm is the earliest lineage to activate the genome" in line 364 – 367.

10. Overall, the methods section should be revised and edited on the grammar level.

R: Thanks for the suggestion, we have revised the Methods section from line 408 to line 558.

Main References:

1. Matsumoto J, *et al.* High-throughput sequence analysis of *Ciona intestinalis* SL trans-spliced mRNAs: alternative expression modes and gene function correlates. *Genome Res.* **20**, 636-645 (2010).
2. Danks GB, *et al.* Trans-splicing and operons in metazoans: translational control in maternally regulated development and recovery from growth arrest. *Mol. Biol. Evo.* **32**, 585-599 (2015).
3. Danks G, Thompson EM. Trans-splicing in metazoans: A link to translational control? *Worm* **4**, e1046030 (2015).
4. Heyn P, *et al.* The earliest transcribed zygotic genes are short, newly evolved, and different across species. *Cell Rep.* **6**, 285-292 (2014).

5. Cao C, *et al.* Comprehensive single-cell transcriptome lineages of a proto-vertebrate. *Nature* **571**, 349-354 (2019).
6. Zhang T, *et al.* A single-cell analysis of the molecular lineage of chordate embryogenesis. *Sci. Adv.* **6**, eabc4773 (2020).
7. Krieg M, *et al.* Tensile forces govern germ-layer organization in zebrafish. *Nat. cell. biol.* **10**, 429-436 (2008).
8. Luxán G, *et al.* Endothelial EphB4 maintains vascular integrity and transport function in adult heart. *Elife* **8**, 45863 (2019).
9. Satou Y, *et al.* A nearly complete genome of *Ciona intestinalis* Type A (*C. robusta*) reveals the contribution of inversion to chromosomal evolution in the genus *Ciona*. *Genome Biol. Evo.* **11**, 3144-3157 (2019).
10. Small KS, Brudno M, Hill MM, Sidow A. A haplome alignment and reference sequence of the highly polymorphic *Ciona savignyi* genome. *Genome Biol.* **8**, R41 (2007).
11. Fujii Y, *et al.* Spatiotemporal dynamics of single cell stiffness in the early developing ascidian chordate embryo. *Commun. Biol.* **4**, 341 (2021).

Reviewer #1 (Remarks to the Author):

The revisions and focusing have significantly improved the manuscript.

The main value of the work is the establishment of *Cirobu/Cisavi* hybrids as a paradigm to study ZGA and ASE, which are understudied in invertebrate chordates.

Whether the manuscript reports an unexpected finding that significantly moves the field forward remains unclear to me.

There are also several sections which are not convincing and the study – and responses to the reviewers' comments – still lacks rigour. There remains quite some way to go before this study can be published in a major journal.

A full review is submitted as supplementary material.

Reviewer #1 Attachment on the following page

REVIEWER COMMENTS

Reviewer #1 (Remarks to the Author) (Text in green corresponds to the second round of assessment)

In this manuscript, Dong and colleagues present a very astute and powerful model system to characterize the maternal to zygotic activation in ascidian embryos: interspecific hybrids between two distantly-related *Ciona* species. They show that these hybrids develop to the tailbud stages and that it is possible to allele-specific dynamics of expression of each gene according to its parent or species of origin. The main strength of the article is the hybrid system used, which is strikingly powerful. The reported experiments analyses are overall sound. Unfortunately, the authors try to cover too much ground, reducing the depth of each analysis. As a consequence, the findings reported are either of a confirmatory nature or insufficiently developed.

Major suggestions:

This article reads like a test run for more interesting studies using *Cirobu/Cisavi* hybrids. It reports many interesting successive snippets of information, each worthy of a more in-depth study. Yet the manuscript currently fails to integrate these snippets into a convincing story. It also lacks a remarkable unexpected result that would make it suitable for a major journal. My main advise would be to choose one of these snippets and dig deeper in the hope to find something unexpected. One possibility would be to analyse more deeply the differences in genetic programs between the two species.

R: Thanks for the comments and suggestions. To improve the integration of the manuscript, we focused on three main findings (temporal hierarchy, spatial dynamics and allele-specific expression of ZGA in *Ciona*), which are logically connected and were revealed using the unique *Ciona* hybrid system. We have reorganized the manuscript. For example, we deleted the gene age part and simplified the gene regulation part in the revision.

In addition, to provide more solid evidences for each aspect of our main findings, we performed additional experiments. For example, we added the CUT&Tag data of H3K27ac and Pol II to support the two waves of ZGA, carried out single cell RNA-Seq of reverse cross hybrid embryos and cell fate transfer experiments to support the new model for ZGA, and conducted motif flip experiments to strengthen the mechanisms of ASE. We hope these additional results will help to integrate the manuscript and strengthen the deeper understanding of our findings.

Regarding to the differences in genetic programs between the two species, we agreed with the reviewer's suggestion. Our results revealed that the species-biased allele specific expression was related to the genetic differences between the two species. We have added the comparative analysis of the activity of cis-regulatory motifs between *C. robusta* and *C. savignyi* by motif flip experiments in the revised manuscript. Nevertheless, regarding to temporal-spatial dynamics of ZGA, the forward and reverse cross hybrids showed similar patterns, indicating that the zygotic genome activation process are largely conserved between the two species. The conservation makes it suitable for using both directional hybrids to study the zygotic genome activation. In the future, study on the genetic difference between the two species using hybrid samples will be an interesting direction.

The revisions and focusing have significantly improved the manuscript. The main value of the work is the establishment of *Cirobu/Cisavi* hybrids as a paradigm to study ZGA and ASE, which are understudied in invertebrate chordates. Whether the manuscript reports an unexpected finding that significantly moves the field forward remains unclear to me. There are also several sections which are not convincing and the study – and responses to the reviewers' comments - still lacks rigour. There is still quite some way to go before this study can be published in a major journal.

Some parts are difficult to follow because of vague or insufficient explanations. This applies in particular to the analysis of the regulatory relationships between maternal and successive waves of zygotic genes (lines 171 to 187). After reading the main text and the materials and methods, I could

still not really understand what had been done and whether it made any sense. I was not even sure that the « motif scanning » section was relevant as the 244 genes described here do not seem to match those of lines 171 and following. Providing a list of the genes being analysed would be a first step. I was also surprised that the authors searched for motifs in the 100-300 bases immediately upstream of what I understood to be the initiator methionine. Many *Ciona* genes are trans-spliced with 5' outrons. The TSS may thus be located some distance away from the initiator methionine (see Matsumoto et al., Genome research, 2010). It would be much preferable to scan open chromatin regions within and upstream of the genes of interest (see Madgwick et al., Dev. Biol. 2019).

R: The TF genes and their motif bearing genes analyzed in this part were used for regulatory relationship analysis. For example, 71 maternal TFs and 301 motif bearing genes can be identified based on JASPAR database. The methods were included in “Regulatory relationship analysis” part in Methods. The TFs in minor waves and their motif bearing genes in major wave have been listed in supplementary Table 4 in the revision. 244 genes were not used for regulatory relationship analysis, but were screened as species-biased genes, which were included in supplementary Table 15. The methods are included in “Motif scanning” part in Methods. Sorry for the unclear statement in the part of regulatory relationships, we have rewritten this part in the revised manuscript (from line 182 to line 191).

I am still quite confused. How confident are the authors in the functional relevance of their “motif bearing genes”? I cannot find in the methods section of the ms how these motif bearing genes were defined (saying Jaspas was used is not enough). My understanding is that TF motifs are usually so degenerate that I expect that any decently long stretch of DNA to have a decent match to a TF binding motif. In addition, there is also considerable evidence that the even more frequent low affinity sites can be functionally relevant. Taken together, I would remove this whole part, except if the authors provide functional evidence that motif bearing genes are indeed direct target genes of the listed TFs.

We agreed with the reviewer that TSS may be located some distance away from the initiator methionine. Therefore, we first conducted upstream sequence (0-1000 bp upstream of initiator methionine) identity analysis on gene homolog pairs. The result revealed significant sequence differences for 100-300 bp sequence window (Extended Data Fig. S22). Therefore, we conducted motif scanning analysis targeting at 100- 300bp region and selected binding sites that may lead to expressional bias.

I still do not understand the logic of the author’s argument. And the legend of Figure S22 is not sufficiently developed for this referee to understand what it shows (is sequence identity really below 50% between these species?). Even if sequence conservation was lower in the 100-300bp segment, why is this a reason for focusing in this part of the sequence. My experience from DNA accessibility analysis is that many accessible regions are more distant than 100-300 bp from the initiator methionine.

Thank you for the indication of trans-splicing events in *Ciona*. It was possible that the TSS were located some distance away from the initiator methionine trans-spliced genes. We checked the spliced leader sequences¹ in our bulk RNA-seq data and found that there was very low ratio (0.001%) of reads containing the spliced leader sequence of *C. robusta*. In addition, it seems that most zygotic mRNA is not trans-spliced in tunicate species according to previous studies in *Oikopleura dioica*^{2,3}. Furthermore, as the reviewer suggested, we provided the CUT&Tag scanning of both Pol II and H3K27Ac, which do help to provide the position of regulatory region of our interested genes (Fig 4c).

0.001% of reads containing the splice leader appears very low. Can the authors compare this to the work of Hastings (In particular Matsumoto et al., 2010)? *Oikopleura* is a very divergent (and odd) tunicate. I am not sure results in *Oikopleura* can be extrapolated to other tunicates.

The results are frequently stated but not interpreted. It is therefore difficult to tell apart mere confirmatory results from true interesting novelties. For instance, do the authors consider the discovery that different classes of genes have different « ages » according to phylostratigraphy as a major result or as a confirmation in *Ciona* of what has been found in other systems?

R: The trend of gene age during ZGA indeed has been discovered in other systems⁴. Here, we showed the similar trend in urochordate using hybrid system. In order to keep the integration of the manuscript, we have deleted this part in the revised manuscript.

Thank you.

Are the genes composing the modules reported line 150 and following orthologous between species? or do the modules consist in different genes in the two species?

R: Because the samples were from hybrid systems, the genes in each module include genes from both species. These genes are composed of both orthologous and different genes between the two species. For $Cr \text{♀} \times Cs \text{♂}$, there are 172 genes from *C. robusta* and 111 genes from *C. savignyi* in module 1 (minor wave), and 98 of 172 and 48 of 111 have orthologous genes between species. There are 1,054 genes from *C. robusta* and 3,301 genes from *C. savignyi* in module 2 and 3 (major wave), and 543 of 1,054 and 2,151 of 3,301 have orthologous genes between species. For $Cs \text{♀} \times Cr \text{♂}$, there are 110 genes from *C. robusta* and 292 genes from *C. savignyi* in module 1 and 39 of 110 and 166 of 292 have orthologous genes between species. There are 1,833 genes from *C. robusta* and 1,264 genes from *C. savignyi* in module 2 and 3, and 1,129 of 1,833 and 696 of 1,264 have orthologous genes between species. These data have been added in supplementary Table 2.

I am still confused. I was not asking whether module genes had orthologs across species, but whether they WERE orthologous between species. Table S3 is not helpful to answer this question. Neither is the text above.

Lines 232-23 : I am not convinced of the claimed lack of correlation between the timing of activation of housekeeping genes and the time elapsed since the last cell division. For instance at the 64-cell stage these genes are preferentially expressed in endoderm and notochord (Fig. 3g) and the precursor cells in these two tissues are the ones that have been around for the longest time (since the 44-cell stage).

R: Here we mean they are not fully supported by the time elapsed since the last cell division. The correlations are indeed suitable for endoderm and notochord cells at the 64-cell stage as the reviewer mentioned, but not for all the cell types such as mesenchyme cells. We have toned down in line 245.

Thank you. I am still confused about Figure S17: why do the authors say that it supports that housekeeping gene activation is not correlated to cell volume or time after last cell division? There is no mention of housekeeping gene expression on this figure.

Minor comments:

Important concepts should be better defined. For instance what is MBT precisely? If it is defined by the desynchronisation of cell divisions, then this occurs in *Ciona* just after the 16-cell stage (see Nishida's early work). Also the 2 « waves » of zygotic transcription that define ZGA should be better defined, and called the same way throughout the ms (currently they are referred to as first and second waves, and as minor and major waves). Also how do the authors define « housekeeping » genes?

R: MBT (mid-blastula transition) were indicated by asynchronous cell divisions and longer cell cycles, and followed by gastrulation. We agreed that the asynchronous cell divisions occurred just

after 16-cell stage in *Ciona*, however, the asynchronous division in *Ciona* was not followed by gastrulation, which indicated a not precise MBT in *Ciona*.

I am not sure why the authors refer to gastrulation when defining MBT. MBT in *Xenopus* occurs hours and many cell divisions before gastrulation. I subscribe to the following definition: “the MBT describes a specific stage during the development of the embryo, which is marked by lengthening and desynchronization of the cell cycles.” (Vastenhouw et al., *Development* (2019) 146 (11): dev161471.)

I disagree that there is no MBT in *Ciona*: The late 16-cell stage seems a perfect match to the classical definition of MBT.

The distinction between ZGA and MBT is an important one, which the authors do not do. For instance, the N/C ratio has been described as a possible mechanism for MBT (NOT for ZGA) in the two references cited (Newport, Jevtič and colleagues),.

For the two waves of ZGA, we should keep the same way throughout the manuscript. We have changed “first wave” and “second wave” into “minor wave” and “major wave” to keep the consistence in the manuscript.

Thank you.

The housekeeping genes were commonly described as essential for cellular existence, and stably expressed irrespective of developmental stage. The housekeeping genes in this analysis were screened by the standard that genes are maternally expressed and have been reactivated at late-neurula stage. We have added the description in Line 204 - 208.

Thank you

Line 92 : *Ciona savignyi* is not « distributed across most coasts around the world » : WORMS indicates that it is mostly found in the North Pacific ocean.

R: The reviewer is right. We have revised this sentence into “Both *C. robusta* and *C. savignyi* are tunicate species with similar morphologies, and are broadly distributed across North Pacific Ocean” in line 101.

Thank you.

While *Halocynthia roretzi* counts 110-cells at the onset of gastrulation, *Ciona* and other phlebobranch species count 112-cells at the same stage (due to the precocious division of the A7.6 cell pair ; see Tassy et al. *Curr. Biol.* 2006 ; Guignard et al. 2020).

R: Thank you for pointing this out. The reviewer is right. We have clarified it in the manuscript “110-cell stage (112 cells at this stage)” in line 415.

“110-cell” stage stands for the initial gastrula stage in *Ciona*. Here we used “110- cell” stage to indicate the cleavage stage in *Ciona*, which is also used in other publications ^{5,6}, not the exact cell number.

I respectfully disagree with the proposed use of “110-cell”, which I ask authors to reconsider. Ascidiaceans are wonderful model organisms because of the incredible precision and invariance of their cell lineage. The 110-cell stage was coined one branch of the ascidian phylogeny, the stolidobranchs. In the *Ciona* branch there are 112 cells at the same developmental stage. Using wrong cell numbers to define a stage is both confusing and inelegant. If the authors do not want to use the correct 112-cell

stage in *Ciona*, they should preclude using cell numbers to stage embryos and use the Hotta developmental stage stages which are conserved across species. That other people are careless in the ascidian community is no excuse.

Lines 74 and 197: The idea that ascidians develop using a mosaic strategy (i.e. one in which cells do not communicate with each other) has been shown wrong by many authors (starting with H. Nishida). The authors are right that ascidian embryos develop with invariant lineages (Conklin, Satoh, Nishida, Christiaen, Lemaire labs among many others), but this is different from « mosaic » development. The term mosaic should be avoided in this context.

R: We agreed with the reviewer that cells during early development of ascidians are not totally independent. Therefore, we avoided using “mosaic” term in the context in revision.

Thank you

Lines 248: Why do the authors describe Nodal and Ephrin genes as « stiffness genes » ? Is there a reference for this?

R: According to the previous reports, Nodal and Ephrin are related to cell stiffness^{7, 8}. They demonstrated the role for Nodal-controlled cell-cortex tension in germ-layer organization and downregulation of Eph results in the decreased cell stiffness. We have added the references in line 254.

I am not sure I agree with the authors: the cited studies use vertebrates, which are more than 500 million years apart from Ascidians. What is the evidence that this function has been conserved across such a vast evolutionary distance?

Lines 289 and Figure 4d : I do not understand why the authors only show binding sites for Fox family genes ?

R: Here We analyzed 86 publicly available motif profiles for motif scanning from JASPAR database, not only for Fox family genes. Since we found the significant sequence differences for upstream 100-300 bp among the 1000 bp (Extended Data Fig. S22). Therefore, we conducted motif scanning analysis targeting at 100-300 bp region and screened the different binding sites. There are indeed many different binding sites from Fox family, but also other binding sites such as otp and Sox family. The binding sites of allelic expression genes were listed in supplementary Table 15 and the diagram for motif distribution was shown in Extended Data Fig. S23.

Restricting this study to the -300 to -100 bp from ATG makes this analysis quite weak. It would make more sense to carry out the analysis on accessible chromatin regions in the vicinity of the genes of interest.

Lines 226-227: I do not understand this sentence.

R: Here we mean housekeeping genes were not expressed simultaneously in each cell during early development. This suggested that ZGA in *Ciona* are not correlated to exact time and did not follow the developmental clock model. We have revised this sentence as “Overall, these results indicated that zygotic housekeeping genes were asynchronized reactivation in hybrid embryos, and thus did not follow the maternal clock model.” in line 238 - 240.

Thank you.

Additional comments:

Title: ascidians are not “basal chordates” (What would Amphioxus be then?) but simply “invertebrate chordates”.

Extended figure S2 panel A: These panels are uninformative because individual embryos cannot be seen because of the scale and low resolution of the picture.

In supplementary table 4, the authors should add the gene model ID of each of the TFs listed, as the name provided is usually not sufficient to find them unambiguously in the field’s Aniseed ascidian database.

Reviewer #3 (Remarks to the Author):

The manuscript is much improved in this revision. The additional experiments conducted and the enhanced data analysis have significantly strengthened the study. Specifically, the inclusion of CUT&Tag of H3K27ac and Pol II data, single cell RNA-seq of reciprocal hybrid embryos, cell fate transfer experiments, and motif flip experiments have provided substantial evidence to support the central findings of the study. Furthermore, the manuscript has been effectively reorganized, focusing on three primary discoveries: temporal hierarchy of ZGA, asynchronized spatial pattern of ZGA, and species-biased allelic gene activation during ZGA. These findings are now more logically connected, bolstered by robust evidence, and contribute to a clearer and more cohesive narrative.

Below are specific points that need to be addressed for acceptance:

1. The low overlap of minor and major wave genes between reciprocal crosses warrants discussion in the text. Please address potential reasons for this discrepancy and discuss its implications.
2. In Supplementary Table 5, "first wave" and "non-first wave" should be changed to "major wave" and "minor wave" for clarity.
3. For previously published data sets, please provide the specific curated gene lists used in this study since the original papers contain multiple tables. This will improve reproducibility.
4. Featurecounts cannot directly generate FPKM values. Please revise the methods to accurately describe FPKM calculation.
5. The results described in lines 232-239 of the revised manuscript refer to the pictures shown in Fig.3g, Extended Data Fig. S16, but to this reviewer, the main text figure and supplemental figure do not look similar at all. Can the authors clarify?
6. Please provide a more in-depth description of the CUT&TAG experimental design and data analysis, to allow the reader the full picture and a way to assess the results.
7. This reviewer acknowledges the great effort of generating the CUT&TAG data from these very early stages. Nonetheless, given the well-known signal pattern of H3K27ac is not observed in the figures, could the authors address this issue in the text?

A point-by-point response

Ref: NCOMMS-23-03866B

Title: Temporospatial hierarchy, dynamics and allele-specific expression of zygotic genome activation in urochordates revealed by distant interspecific hybrids

REVIEWER COMMENTS

Reviewer #1 (Remarks to the Author):

The revisions and focusing have significantly improved the manuscript. The main value of the work is the establishment of *Cirobu/Cisavi* hybrids as a paradigm to study ZGA and ASE, which are understudied in invertebrate chordates. Whether the manuscript reports an unexpected finding that significantly moves the field forward remains unclear to me. There are also several sections which are not convincing and the study – and responses to the reviewers’ comments - still lacks rigour. There is still quite some way to go before this study can be published in a major journal.

I am still quite confused. How confident are the authors in the functional relevance of their “motif bearing genes”? I cannot find in the methods section of the ms how these motif bearing genes were defined (saying Jaspar was used is not enough). My understanding is that TF motifs are usually so degenerate that I expect that any decently long stretch of DNA to have a decent match to a TF binding motif. In addition, there is also considerable evidence that the even more frequent low affinity sites can be functionally relevant. Taken together, I would remove this whole part, except if the authors provide functional evidence that motif bearing genes are indeed direct target genes of the listed TFs.

R: In this part, we identified these motif bearing genes by searching their upstream sequences for matches of motif sequences in JASPAR database (<https://jaspar.elixir.no/>, 2022). The motif bearing genes are regarded as possible downstream genes of the corresponding transcription factors. In agreement with the reviewer, we recognize that simple searching for motif sequences in JASPAR may not be entirely convincing and lack of functional evidences. To improve the integration of the manuscript, we have deleted this part as the reviewer suggested. Supplementary Table 4 - 5 and the extended data Fig. S7 were removed as well in the revision.

I still do not understand the logic of the author's argument. And the legend of Figure S22 is not sufficiently developed for this referee to understand what it shows (is sequence identity really below 50% between these species?). Even if sequence conservation was lower in the 100-300bp segment, why is this a reason for focusing in this part of the sequence. My experience from DNA accessibility analysis is that many accessible regions are more distant than 100-300 bp from the initiator methionine.

R: Fig. S22 (Fig. S21 in revision) showed the upstream sequence identities between *C. robusta* and *C. savignyi*. The 1,000 bp upstream sequences were divided every 100 bp, and the gene pairs were divided into species-biased, parental-biased, and BUSCO genes, respectively. The upstream sequences of each gene pair were aligned using BLASTN. The identity percent of each 100 bp were shown in the figure. We have added the figure legend for Fig. S22 (Fig. S21 in revision). The average sequence identities between the two species were indeed below 50%, and we inferred that this is because they are upstream non-coding regions, and sequence identities of these regions are indeed lower than those of gene coding regions.

We agreed with the reviewer that “many accessible regions are more distant than 100-300 bp from the initiator methionine”. When conducting this analysis, our primary objective was to distinguish the difference between the upstream regions of parent-biased genes and species-biased genes. In essence, were the upstream regions of species-biased genes more divergent than those of parent-biased genes? Interestingly, only parts of the upstream region (100 -300 bps) were significant divergent. This is the reason why we focus on 100-300 bps. The sequence identity of upstream 100 - 300 bps was significantly lower (p -value = 0.04 in 100 - 200 bps, p -value = 0.02 in 200 - 300 bps) in species-biased gene pairs than in parent-biased gene pairs. The sequence similarities were related to the conservation of cis-regulatory elements to some extent^{1, 2}, we thus inferred that the different cis-regulatory elements were enriched in these divergent regions between species-biased gene pairs. The following motif distribution analysis indeed revealed that there existed different cis-regulatory elements in this region (Fig. S23, now is Fig. S22 in revision). We have added the above description and interpretation in the revision in Line 303-304.

As the reviewer suggested, the analysis on accessible chromatin regions in the vicinity of the genes of interest is indeed a good way. ATAC-seq is widely used to

identify open chromatin regions. In order to analyze the accessible chromatin regions, we downloaded and analyzed the ATAC-seq datasets of 64-cell and 112-cell stage from *C. robusta* (Bioproject PRJNA474983)³ using R package “ChIPseeker”⁴. We identified 11041, 14214, 7672, and 11129 peaks for each sample. The peak distribution of *C. robusta* at 64-cell and 112-cell stage was shown in Table R1. We found that about 62.6% of peaks are distributed in promoter region. However, there is still a lack of ATAC-seq data at early developmental stages of *C. savignyi*. We cannot compare the accessible chromatin regions of each gene pair from the two species directly. If we acquired the ATAC-seq data from both species, it can help to indicate the active regulatory elements. But even though, to identify the differences of the motifs between the two species, we still need to compare the sequence identity and match them with motif databases. In the present analysis, we cannot make sure all of the cis-regulatory elements are included, but the variation in these regions also could help to reveal the regulatory differences and allele-specific expression in *Ciona*.

In order to validate the analysis about the variation of cis-regulatory elements between the two species using 1 kb and then focus on -300 to -100 bp from initiator methionine still make sense to some extent, we supplied the following analysis:

Firstly, in order to analyze the distribution of possible active regulatory elements, we downloaded and analyzed the ATAC-seq datasets (Bioproject PRJNA474983) at 64-cell and 112-cell stage of *C. robusta* from previous study in embryonic cis-regulatory landscapes of *Ciona*³. For these peaks in promoter region, 72.6% of the peaks were within 1 kb of transcription start site (TSS), 61.7% of the peaks were within upstream 1 kb of initiator methionine, 28.1% were distributed in upstream 100 - 300 bps from initiator methionine, which indicated that upstream regions of initiator methionine really enriched many cis-regulatory elements.

Table R1. The ATAC-seq peak distribution of *C. robusta* at 64-cell and 112-cell stages

Peak distribution	64-cell-1	64-cell-2	112-cell-1	112-cell-2
Total	11041	14214	7672	11129
Promoter	7077	8632	4870	7000
Promoter (tss<=1kb)	5160	6153	3559	5151
Promoter (tss>1kb)	1917	2479	1311	1849
Exon	1737	3101	1125	1782

Distal Intergenic	885	980	634	927
Intron	1316	1470	1028	1400
Downstream	26	31	15	20
upstream 1kb of initiator methionine	3117	3587	2307	3282
upstream 100 – 300 bp of initiator methionine	1478	1567	1103	1524

Secondly, when we calculated the distance between initiator methionine and transcription start site (TSS) (i.e. the length of 5'UTR), we found that 86.4% of genes from *C. robusta* and 72.3% of genes from *C. savignyi* are within 100 bps (Table R2). This referred that the 100 – 300 bps upstream of initiator methionine would also cover the upstream regions of TSS for most of the genes. The reason why we chose upstream sequences from the initiator methionine not from the TSS is because the *Ciona* genome is compact and the annotation of TSS in *Ciona* genome is not as good as other model organisms. In addition, when we analyzed the sequence identity of 1,000 bps upstream and 500 bps downstream of the TSS in different gene pairs between *C. robusta* and *C. savignyi*, we found a significantly lower region (p -value = 0.0089 for upstream 100 - 200 bp) in species-biased gene pairs than in maternal-biased gene pairs (Figure R1), which is similar with the analysis using upstream region of initiator methionine.

Table R2. The statistics of distance between initiator methionine and TSS (i.e. the length of 5'UTR) of *C. robusta* and *C. savignyi*

5'UTR length	median	mean	<100 bp
C. robusta	26	157.4755	15712
C. savignyi	37	285.0573	8401

Figure R1. The sequence identity of 1,000 bps upstream and 500 bps downstream of the TSS in different gene pairs between *C. robusta* and *C. savignyi*. The sequences were divided every 100 bps, and the gene pairs were divided into species biased, parental biased, and BUSCO genes. The sequences of each gene pair were aligned using BLASTN. The identity percent of each 100 bps were shown. The *p*-value of sequence identity between species-biased group and other groups were labeled.

Considering the above, we admitted that restricting to the -300 to -100 bp from initiator methionine was not rigorous enough for cis-regulatory analysis. We have toned down in the revised manuscript in Line 397-401. Hopefully we could perform more functional experiments in the future. Nevertheless, we think the present analysis about the variation of cis-regulatory elements between the two species using 1 kb and then focus on -300 to -100 bp from initiator methionine still make sense to some extent. The following H3K27ac and Pol II occupancy analysis (Fig. 4c) and promoter-replace validation experiments (Fig. 4d and Fig. S23) in this region also support our conclusion.

0.001% of reads containing the splice leader appears very low. Can the authors compare this to the work of Hastings (In particular Matsumoto et al., 2010)? *Oikopleura* is a very divergent (and odd) tunicate. I am not sure results in *Oikopleura* can be extrapolated to other tunicates.

R: The work of Hastings⁵ identified 8790 SL trans-spliced mRNAs using SL-PCR-amplified random-primed reverse transcripts of tailbud embryo RNA, which significantly enriched the possible SL trans-splicing events. Their purpose is to acquire all the possible existence of SL trans-spliced genes but not to evaluate the efficiency of trans-splicing events during early development. Since our RNA-seq libraries were not constructed by the specific primers, we think that the ratio of splice leader sequences cannot be compared directly. Nevertheless, 0.001% was indeed very low. We realized that the bulk RNA-seq data would underestimate the ratio of splice leader (SL) trans-splicing events because its low efficiency in capturing 5' sequences of mRNA.

In order to check the SL sequence ratio in Next Generation Sequencing RNA-seq data, we downloaded another bulk RNA-seq data “RNA-Seq data of *Ciona intestinalis*: Whole Embryo WT rep1 Stage 11 (Early Gastrula) (SRR6283060)” from NCBI, and searched the SL sequence. The results showed that 0.0028% of reads contained the splice leader sequence, which is similar with our hybrid RNA-seq data.

We understand the reviewer’s concern that *Oikopleura* is a divergent tunicate, the situation in *Oikopleura* cannot represent that in *Ciona*. Since they retained the trans-splicing events, it might be used as a reference to infer the situation in other tunicates to some extent. Hope we can perform more experiments in the future to figure it out.

Considering the above, we admitted that the trans-splicing events would influence the identification of the locations of gene regulatory elements. We thus have toned down and added the discussion regarding trans-splicing in Discussion part in Line 397-401.

Nevertheless, according to the response to the previous comment, we think the present analysis about the variation of cis-regulatory elements between the two species still make sense to some extent. We proposed one possible mechanism regarding the allele-specific expression in this study. The other potential mechanisms will be addressed in additional following up work.

I am still confused. I was not asking whether module genes had orthologs across species, but whether they WERE orthologous between species. Table S3 is not helpful to answer this question. Neither is the text above.

R: We compared the paternal and maternal genes in each module. For Cr ♀ × Cs ♂, there were 14 orthologous gene pairs in module 1, and 433 orthologous gene pairs in module 2 and 3 between species. For Cs ♀ × Cr ♂, there were 9 orthologous gene pairs in module 1, and 353 orthologous gene pairs in module 2 and 3 between species. We have added these data in supplementary Table 3.

Thank you. I am still confused about Figure S17: why do the authors say that it supports that housekeeping gene activation is not correlated to cell volume or time after last cell division? There is no mention of housekeeping gene expression on this figure.

R: The housekeeping gene expression at 64-cell and 112-cell stages were shown in Fig. 3c and f. We supplied the data in Fig. S17 (Fig. S16 in revision) in the revised manuscript to compare the trends directly. When we compared the trends in different cell types, we could find that the housekeeping gene activation was not exactly correlated to cell volume or time after last cell division at both 64-cell and 112-cell stages. For example, the housekeeping gene activation is correlated to cell volume or time after last cell division in endoderm and notochord cells at 64-cell stage, but not at 112-cell stage.

I am not sure why the authors refer to gastrulation when defining MBT. MBT in *Xenopus* occurs hours and many cell divisions before gastrulation. I subscribe to the following definition: “the MBT describes a specific stage during the development of the embryo, which is marked by lengthening and desynchronization of the cell cycles.” (Vastenhouw et al., *Development* (2019) 146 (11): dev161471.) I disagree that there is no MBT in *Ciona*: The late 16-cell stage seems a perfect match to the classical

definition of MBT. The distinction between ZGA and MBT is an important one, which the authors do not do. For instance, the N/C ratio has been described as a possible mechanism for MBT (NOT for ZGA) in the two references cited (Newport, Jevtič and colleagues).

R: Thanks for the comments. We agreed with the reviewer. Actually, we did not pay too much attention on the distinction between ZGA and MBT in our manuscript. We have deleted the statement about MBT and revised the sentence into “the regulation of ZGA in non-model animals remains poorly understood.” in Line 67 in the revised manuscript.

I respectfully disagree with the proposed use of “110-cell”, which I ask authors to reconsider. Ascidiaceans are wonderful model organisms because of the incredible precision and invariance of their cell lineage. The 110-cell stage was coined one branch of the ascidian phylogeny, the stolidobranchs.

In the *Ciona* branch there are 112 cells at the same developmental stage. Using wrong cell numbers to define a stage is both confusing and inelegant. If the authors do not want to use the correct 112-cell stage in *Ciona*, they should preclude using cell numbers to stage embryos and use the Hotta developmental stage stages which are conserved across species. That other people are careless in the ascidian community is no excuse.

R: Thanks for the comments. The reviewer is right. In order to avoid the misunderstanding, we have changed all the “110-cell” stage into “112-cell” stage in the text and figures in the revised manuscript.

I am not sure I agree with the authors: the cited studies use vertebrates, which are more than 500 million years apart from Ascidiaceans. What is the evidence that this function has been conserved across such a vast evolutionary distance?

R: The gene functions were indeed not tested in *Ciona*. However, they were orthologs with genes in vertebrates. Nodal in *C. robusta* patterns the neural plate along the medial–lateral axis ⁶ and is also required for induction of muscle and notochord cell fates ⁷. Ephrin signaling drives the asymmetric division of notochord/neural precursors ⁸ and endomesoderm lineage of the *Ciona* embryo⁹. These results indicated that *nodal* and *ephrin* genes in ascidiaceans have cell communication function, which is similar with that in vertebrates. Then we speculated that they also have similar cell stiffness related functions as in vertebrates. The determined function of these stiffness-related signaling genes in ascidiaceans are needed to validate in the future.

Restricting this study to the -300 to -100 bp from ATG makes this analysis quite weak. It would make more sense to carry out the analysis on accessible chromatin regions in the vicinity of the genes of interest.

R: The analysis on accessible chromatin regions in the vicinity of the genes of interest is indeed a good way. Hopefully the experiments could be performed in the future. We explained this in previous response (from page 2 to 6). Please refer to the above.

Additional comments:

Title: ascidians are not “basal chordates” (What would Amphioxus be then?) but simply “invertebrate chordates”.

R: Thanks for the suggestions. We have changed the basal chordates into urochordates.

Extended figure S2 panel A: These panels are uninformative because individual embryos cannot be seen because of the scale and low resolution of the picture.

R: Sorry for the scale and low resolution in Fig. S2. We now zoomed in and showed the individual embryos at each stage in the revision.

In supplementary table 4, the authors should add the gene model ID of each of the TFs listed, as the name provided is usually not sufficient to find them unambiguously in the field’s Aniseed ascidian database.

R: Thanks for the suggestions. We have deleted this part in the revised manuscript. This supplementary table was also removed.

Reviewer #3 (Remarks to the Author):

The manuscript is much improved in this revision. The additional experiments conducted and the enhanced data analysis have significantly strengthened the study. Specifically, the inclusion of CUT&Tag of H3K27ac and Pol II data, single cell RNA-seq of reciprocal hybrid embryos, cell fate transfer experiments, and motif flip experiments have provided substantial evidence to support the central findings of the study. Furthermore, the manuscript has been effectively reorganized, focusing on three primary discoveries: temporal hierarchy of ZGA, asynchronized spatial pattern of ZGA, and species-biased allelic gene activation during ZGA. These findings are now more logically connected, bolstered by robust evidence, and contribute to a clearer and more cohesive narrative.

Below are specific points that need to be addressed for acceptance:

1. The low overlap of minor and major wave genes between reciprocal crosses warrants discussion in the text. Please address potential reasons for this discrepancy and discuss its implications.

R: Thanks for the comments and suggestions. The overlap of minor and major wave genes between reciprocal crosses was indeed low according to our data. The potential reason might be related to the deeply divergence between the two species (120 Mya). There are 18,191 genes annotated on the reference genome of *C. robusta* and 12,172 genes annotated on the reference genome of *C. savignyi*. Only 5,879 orthologous gene pairs were detected using bi-direction BLAST analysis. However, when we conducted the GO enrichment analysis of minor and major wave genes, respectively, the two reciprocal crosses showed similar results (Fig. S6). This indicated that the general patterns of possible regulation relationships in the two hybrids were similar.

2. In Supplementary Table 5, "first wave" and "non-first wave" should be changed to "major wave" and "minor wave" for clarity.

R: Thanks for the reminding. We should revise them in supplementary table 5. However, according to reviewer 1's suggestion, we have deleted this table in the revised manuscript.

3. For previously published data sets, please provide the specific curated gene lists used in this study since the original papers contain multiple tables. This will improve reproducibility.

R: Thanks for the suggestions. The genes in *C. robusta* were from Ghost (KY2019, <http://ghost.zool.kyoto-u.ac.jp/default.html>) and genes in *C. savignyi* were from Ensembl (CSAV2.0, <https://asia.ensembl.org/index.html>). The gene lists used in this study have been added in supplementary table 2.

4. Featurecounts cannot directly generate FPKM values. Please revise the methods to accurately describe FPKM calculation.

R: Thanks for the suggestions. FeatureCounts were used to acquire the reads counts mapped to each gene and then to acquire the FPKM value of each gene. FPKM of each gene was then calculated based on the length of the gene and reads count mapped to this gene¹⁰. We have revised it in the manuscript in Line 437-440.

5. The results described in lines 232-239 of the revised manuscript refer to the pictures shown in Fig.3g, Extended Data Fig. S16, but to this reviewer, the main text figure and supplemental figure do not look similar at all. Can the authors clarify?

R: Sorry for the insufficient description. We conducted the whole-mount *in situ* hybridization experiments of 60SL11, RPL13, SEC, KRR and HNRN in hybrid *Ciona* (Cs♂ × Cr♀) and self-crossed *C. savignyi* embryos at 32-cell, 64-cell and 112-cell stages. The results of anti-sense probes were shown in Fig. 3g, while the results of sense probes were shown in Fig. S16 as control groups (Now is Fig. S15 in revision). The anti-sense probes could be hybridized with targeted RNA, while the sense probes cannot. There were no signals when using sense probes (shown in Fig. S16, Now is Fig. S15 in revision), which indicated the signals acquired from anti-sense probes were specific gene expression signals (shown in Fig. 3g). We have added the description in the revised manuscript in Line 232-233 and Fig. S15.

6. Please provide a more in-depth description of the CUT&TAG experimental design and data analysis, to allow the reader the full picture and a way to assess the results.

R: Thanks for the suggestions. We have added the description about the experimental design and data analysis about CUT&Tag in both results and methods in Line 159-161, 289-298, and Line 448-454, respectively.

“The CUT&Tag data can provide efficient chromatin features to indicate nascent gene expression, which is a complementary to RNA-seq analysis.”

“To validate the biased expression features during zygotic genome activation, the CUT&Tag results were visualized on the biased-expressed genes. The signal pattern of H3K27ac was weak at early stages. The resulting curves and heatmaps still showed the relative signals for genes with expression-biased features (Extended Data Fig. S20). Under most circumstances, for *C. robusta* biased genes, signals mapped on *C. robusta* genome have higher peak in both results targeting H3K27ac and RNA pol II, implying a dominant characteristic for transcription suites from *C. robusta*. For *C. savignyi* biased genes, signals mapped on *C. savignyi* genome showed an improved trend compared to that on *C. robusta* genome, which is consistent with the activation of respected genes. In maternal-based genes, a similar trend was also observed.”

“The raw data were split with python tool barcode splitter (version 0.18.6) according to sequencing barcodes. The fastq files were aligned against a combined reference genome of *C. robusta* and *C. savignyi* using Bowtie2 version 2.4.4 to produce Bam files that were transferred to browsable BW file using bamCoverage version 3.5.1. The computeMatrix (version: 3.5.1) scale-regions function was used to compute the signal. The plotHeatmap was used to visualize the results with respected BED files of genes”.

7. This reviewer acknowledges the great effort of generating the CUT&TAG data from these very early stages. Nonetheless, given the well-known signal pattern of H3K27ac is not observed in the figures, could the authors address this issue in the text?

R: Thanks for your comments. We admitted that the canonical signal patterns of H3K27ac at early stages were not obvious. We inferred that one of the reasons is that the transcribed gene numbers are very few at early stages. Along with the zygotic gene expression, the signal patterns could be observed and more obvious at 112-cell stage. In addition, the genome annotation would also influence the results. The precious annotation of TSS position is crucial to acquire the canonical signal patterns of H3K27ac. We think that although there is a lack of canonical signal patterns of H3K27ac at early stages, the CUT&Tag data still could help to support the gene expression patterns during zygotic genome activation of *Ciona*. We have added the description in Line 290-292.

References

1. Maeso I, Irimia M, Tena JJ, Casares F, Gómez-Skarmeta JL. Deep conservation of cis-regulatory elements in metazoans. *Philos. Trans. R. Soc. B-Biol. Sci.* **368**, 20130020 (2013).
2. Clarke SL, VanderMeer JE, Wenger AM, Schaar BT, Ahituv N, Bejerano G. Human developmental enhancers conserved between deuterostomes and protostomes. *PLoS Genet.* **8**, e1002852 (2012).
3. Madgwick A, *et al.* Evolution of embryonic cis-regulatory landscapes between divergent *Phallusia* and *Ciona* ascidians. *Dev. Biol.* **448**, 71-87 (2019).
4. Yu G, Wang LG, He QY. ChIPseeker: an R/Bioconductor package for ChIP peak annotation, comparison and visualization. *Bioinformatics* **31**, 2382-2383 (2015).
5. Matsumoto J, *et al.* High-throughput sequence analysis of *Ciona intestinalis* SL trans-spliced mRNAs: Alternative expression modes and gene function correlates. *Genome Res.* **20**, 636-645 (2010).
6. Hudson C, Yasuo H. Patterning across the ascidian neural plate by lateral Nodal signalling sources. *Development* **132**, 1199-1210 (2005).
7. Hudson C, Yasuo H. A signalling relay involving Nodal and Delta ligands acts during secondary notochord induction in *Ciona* embryos. *Development* **133**, 2855-2864 (2006).
8. Picco V, Hudson C, Yasuo H. Ephrin-Eph signalling drives the asymmetric division of notochord/neural precursors in *Ciona* embryos. *Development* **134**, 1491-1497 (2007).
9. Shi W, Levine M. Ephrin signaling establishes asymmetric cell fates in an endomesoderm lineage of the *Ciona* embryo. *Development* **135**, 931-940 (2008).
10. Trapnell C, *et al.* Transcript assembly and quantification by RNA-Seq reveals unannotated transcripts and isoform switching during cell differentiation. *Nat. Biotechnol.* **28**, 511-515 (2010).

Reviewer #1 (Remarks to the Author):

The new revision carried out by the authors has again improved the quality of this manuscript. I continue to think the manuscript has a potential to be published in a major journal because of the originality of the approach and the importance of the topic. However, as detailed below, I am still very hesitant to recommend publication, as the authors continue to overinterpret some of their data, and provide insufficient details to really understand what has been done.

The strengths of the manuscript are: 1) the hybrid system used that allow to track ZGA by monitoring paternal gene expression; 2) the demonstration that ZGA is asynchronous in cells of distinct fates and uncorrelated to cell volume and cell cycle.

Conversely, the sections that are weakest and, in my opinion, should be excluded from the manuscript are: 1) the relationships between stiffness and ZGA; 2) the cis-regulatory analysis.

I have difficulties sufficiently understanding the part concerning species-specific gene activation to assess it, but this may be more because the text is very difficult to follow than because the data are unsupportive to the claim. This may reflect a more general problem with the editorial quality of the manuscript.

Major issues:

1) Data shown in some figures are still not supporting the text (only 2 examples are provided, but the authors should critically review all of their figures in main and supplemental data for similar issues).

a. Fig 3b does NOT show that mesenchyme and germline have the lowest paternal transcript ratio at the 64-cell stage. I would rather say that muscle and Neural do (but this may not be statistically significant, and no stats are provided).

b. Fig 3h is probably irrelevant to the argument that stiffness plays a role in ZGA. Even if one believes the stiffness measurements provided, they would only make some sense if the experiment was carried out at the 64- to 112-cell stage. Although no developmental stage is mentioned in the panel legend, the small size of cells strongly suggests that the embryos are much beyond these stages, making the analysis irrelevant. It should be done again at the right stage or removed from the manuscript.

2) Data in some figures are impossible to trace (again these are only examples that suggest a systematic check should be done by authors – and Nature comms editor).

a. Extended data Fig S17a for instance shows measurements of stiffness. But there is nowhere in the manuscript (text, legend, methods section) any description of the origin of the data shown.

b. Extended data Fig. 18a-c are very confusing to me. The two axes are supposed to show a ratio of gene expression levels (i.e. positive values) but the values go from -10 to +10. Is it because log values are shown? If so, how can there be such small values for these ratios, which I would expect to be more or less always superior to 1 (before applying the log)? Even if it was, I cannot understand the interpretation of the panels. To me, the only thing they show is that the data tend to be correlated along the two axes (i.e. they align along a diagonal). The apparent difference between stages is difficult to interpret as the number of dots differ, and there are no statistical analyses to say whether behaviours actually statistically differ. But even if they did, I completely fail to see why the authors say that these panels support that “allelic differential activation in hybrid embryos tended to be species- rather than parental-manner (sic) during major ZGA”. If it was, wouldn't one expect that the dots align along a vertical line rather than a diagonal one? I may just have missed something, but I suspect that many readers would as well.

3) Cis-regulatory element analysis is still confusing to me.

a. A correlation between sequence conservation and functional conservation of cis-regulatory elements may be true in some contexts. But it is neither true in flies (see 10.1371/journal.pgen.1000106) nor in ascidians (see ref 42 of the submitted manuscript). Such a correlation also runs against the idea that there can be considerable neutral TF binding site turnover in cis-regulatory elements. The starting point of this analysis is thus very weak.

b. The experimental test of the FoxB TFBS swap in the titin gene is superficially suggestive, but it

assumes, without showing, that this site is actually important for the activity of the Cr titin cis-element. Other TFs may bind at the same location. As I could not find a sufficient description of the mutations the authors introduced in the flip experiments (can this be extracted from table S14? And how?), this experiment is impossible to reproduce and to interpret.

4) The authors should considerably improve the clarity of the description of their experiments. In many cases I really struggled to understand what was meant, and I am not sure I always managed. A few – of many, many examples –

a. "We inferred that these species-of-origin effect allele-genes in hybrid Ciona were related to variation in differential cis-regulatory elements ». I do not understand "species-of-origin effect allele-genes".

b. "we compared the cell stiffness according to the expression of cell types". What is the "expression of cell types"?

Overall, the authors should be praised for the beauty and potential of the system they are reporting, but their low level of rigour and of editorial skills undermines the work. It is a real pity. Please make a real effort to make this manuscript reach sufficient scientific standards! Your work is worth it!

Reviewer #3 (Remarks to the Author):

Authors have responded satisfactorily to the points raised, while toning down some of their original claims, therefore I am now in support of publication.

A point-by-point response

Ref: NCOMMS-23-03866B

Title: Temporospatial hierarchy, dynamics and allele-specific expression of zygotic genome activation in urochordates revealed by distant interspecific hybrids

REVIEWER COMMENTS

Reviewer #1 (Remarks to the Author):

The new revision carried out by the authors has again improved the quality of this manuscript. I continue to think the manuscript has a potential to be published in a major journal because of the originality of the approach and the importance of the topic. However, as detailed below, I am still very hesitant to recommend publication, as the authors continue to overinterpret some of their data, and provide insufficient details to really understand what has been done.

The strengths of the manuscript are: 1) the hybrid system used that allow to track ZGA by monitoring paternal gene expression; 2) the demonstration that ZGA is asynchronous in cells of distinct fates and uncorrelated to cell volume and cell cycle.

Conversely, the sections that are weakest and, in my opinion, should be excluded from the manuscript are: 1) the relationships between stiffness and ZGA; 2) the cis-regulatory analysis.

I have difficulties sufficiently understanding the part concerning species-specific gene activation to assess it, but this may be more because the text is very difficult to follow than because the data are unsupportive to the claim. This may reflect a more general problem with the editorial quality of the manuscript.

Major issues:

1) Data shown in some figures are still not supporting the text (only 2 examples are provided, but the authors should critically review all of their figures in main and supplemental data for similar issues).

a. Fig 3b does NOT show that mesenchyme and germline have the lowest paternal transcript ratio at the 64-cell stage. I would rather say that muscle and Neural do (but this may not be statistically significant, and no stats are provided).

b. Fig 3h is probably irrelevant to the argument that stiffness plays a role in ZGA. Even if one believes the stiffness measurements provided, they would only make some sense if the experiment was carried out at the 64- to 112-cell stage. Although no developmental stage is mentioned in the panel legend, the small size of cells strongly suggests that the embryos are much beyond these stages, making the analysis irrelevant. It should be done again at the right stage or removed from the manuscript.

R:

a. Fig. 3b. There is indeed a lack of statistics on the lowest paternal transcript ratio at the 64-cell stage. We therefore deleted the statement in the text in Line 213. We have further checked other figures to make them clearly supporting the text.

b. Fig. 3h. We admitted that it would be much better to make the stiffness measurements at the 64- to 112-cell stages as the reviewer indicated. However, the vinculin tension sensor that we used in this experiment is based on the FRET, in which the signal intensity needs to be sufficient strong for imaging. The vinTS tension sensor expression was observed at each developmental stage (including 64-cell and 112-cell stage) after addition of inhibitor at 16-cell stage until the signal level became high sufficient for imaging and measurement. Based on our observation, the earliest time point when we could acquire the sufficient signal was mid-tailbud stage. We inferred that although the FoxA promoter drives the tension sensor expression at 64- to 112-cell stage, the fluorescence proteins still need a longer time to accumulate. Considering the measurement was carried out beyond the 64-cell and 112-cell stage and the results only suggested a correlation between cell fates and cell stiffness, we have moved this part to the supplementary Fig. S17c and toned down our statement as well in Line 259 - 260.

2) Data in some figures are impossible to trace (again these are only examples that suggest a systematic check should be done by authors – and Nature comms editor).

a. Extended data Fig S17a for instance shows measurements of stiffness. But there is nowhere in the manuscript (text, legend, methods section) any description of the origin of the data shown.

b. Extended data Fig. 18a-c are very confusing to me. The two axes are supposed to show a ratio of gene expression levels (i.e. positive values) but the values go from -10 to +10. Is it because log values are shown? If so, how can there be such small values for these ratios, which I would expect to be more or less always superior to 1 (before applying the log)? Even if it was, I cannot understand the interpretation of the panels. To me, the only thing they show is that the data tend to be correlated along the two axes (i.e. they align along a diagonal). The apparent difference between stages is difficult to interpret as the number of dots differ, and there are no statistical analyses to say whether behaviours actually statistically differ. But even if they did, I completely fail to see why the authors say that these panels support that “allelic differential activation in hybrid embryos tended to be species- rather than parental-manner (sic) during major ZGA”. If it was, wouldn't one expect that the dots align along a vertical line rather than a diagonal one? I may just have missed something, but I suspect that many readers would as well.

R:

a. Extended data Fig. S17a. The origin of the data was acquired from the previous study¹. The stiffness of cells in ascidian embryos before gastrulation were investigated using atomic force microscopy. We cited the reference and listed the data and reference in supplementary Table 11. Clearly, we indeed should also state in the main text. We thus have added the data source description in Line 242 – 243.

b. Extended data Fig. S18a-c, the two axes are indeed supposed to show the ratio of gene expression levels. The values go from -10 to +10, because they are transformed by \log_2 . The X axis indicated the \log_2 transformed expression ratio between genes from *C. robusta* and the allelic genes from *C. savignyi* in reverse cross ($Cs_{\text{♀}} \times Cr_{\text{♂}}$) embryos. When the ratio is 1, the transformed value is 0; When the ratio is larger than 1, the transformed value is a positive number; When the ratio is smaller than 1, the transformed value is a negative number. Similarly, the Y axis indicated the \log_2 transformed expression ratio between genes from *C. robusta* and the allelic genes from *C. savignyi* in forward cross ($Cr_{\text{♀}} \times Cs_{\text{♂}}$) embryos. Therefore, the first quadrant indicated the expression level of gene from *C. robusta* was higher than its allelic gene from *C. savignyi* in both forward and reverse crosses, then we define it as *C. robusta*

dominant. Similarly, the second quadrants indicated gene expression level was maternal dominant in both forward and reverse crosses, the third quadrant was *C. savignyi* dominant in both crosses and the fourth quadrant was paternal dominant in both crosses. The data align along a diagonal line indicated the gene expression ratios between gene from *C. robusta* and the allelic gene from *C. savignyi* in forward cross and reverse cross were similar. There are more dots in the first and third quadrants than second and fourth quadrants, then we inferred more genes are species-biased rather than parental-biased. Sorry for the insufficient description, we have added above information in Fig. S18 legend to make it clear for the audience.

3) Cis-regulatory element analysis is still confusing to me.

a. A correlation between sequence conservation and functional conservation of cis-regulatory elements may be true in some contexts. But it neither true in flies (see 10.1371/journal.pgen.1000106) nor in ascidians (see ref 42 of the submitted manuscript). Such a correlation also runs against the idea that there can be considerable neutral TF binding site turnover in cis-regulatory elements. The starting point of this analysis is thus very weak.

b. The experimental test of the FoxB TFBS swap in the titin gene is superficially suggestive, but it assumes, without showing, that this site is actually important for the activity of the Cr titin cis-element. Other TFs may bind at the same location. As I could not find a sufficient description of the mutations the authors introduced in the flip experiments (can this be extracted from table S14? And how?), this experiment is impossible to reproduce and to interpret.

R:

a. We agreed that the sequence conservation of cis-regulatory elements was not equal to the functional conservation. Here, we only provided one possible interpretation contributing to the expression bias in the hybrid system and tried to address the potential mechanisms from the cis-regulatory element.

b. We are sorry for the insufficient description of the TFBS swap experiment. We added details in the Result section (Line 326 - 330), and Material and methods section (Line 565 - 566). We also added a new table (Supplementary Table 14) to provide the sequences for the promoter region which were swapped. In addition, we agreed with

the reviewer that this site may not be FoxB specific, we have toned down the statement on binding site in the legends of Fig. 4.

4) The authors should considerably improve the clarity of the description of their experiments. In many cases I really struggled to understand what was meant, and I am not sure I always managed. A few – of many, many examples –

a. “We inferred that these species-of-origin effect allele-genes in hybrid *Ciona* were related to variation in differential cis-regulatory elements ». I do not understand “species-of-origin effect allele-genes”.

b. “we compared the cell stiffness according to the expression of cell types”. What is the “expression of cell types”?

R: Thanks for the comments.

a, we mean genes which showed species-biased expression for “species-of-origin effect allele-genes”. We have revised the sentence into “We inferred that genes that showed species-biased expression in hybrid *Ciona* might be due to variations of cis-regulatory elements.” in Line 301 – 302.

b, we mean that we compared the cell stiffness and paternal housekeeping gene expression in different cell types. We have revised this sentence in Line 245.

In order to considerably improve the clarity of the description, we also asked the professional language editing agency to polish the whole manuscript.

Overall, the authors should be praised for the beauty and potential of the system they are reporting, but their low level of rigour and of editorial skills undermines the work. It is a real pity. Please make a real effort to make this manuscript reach sufficient scientific standards! Your work is worth it!

R: We really appreciate the reviewer for all the comments and suggestions, which are insights and great helpful on the improvement of our manuscript.

Reviewer #3 (Remarks to the Author):

Authors have responded satisfactorily to the points raised, while toning down some of their original claims, therefore I am now in support of publication.

R: Thanks for the comments. We really appreciate the comments and suggestions from the reviewer.

Reference

1. Fujii Y, *et al.* Spatiotemporal dynamics of single cell stiffness in the early developing ascidian chordate embryo. *Communications Biology* **4**, (2021).

Reviewer #1 (Remarks to the Author):

The authors have now done a great deal to satisfy my concerns. The text may still a bit of polishing by the Nature communication editorial staff, but I now support publication of this study in the journal and look forward to seeing the article published.

I apologise to the authors for having insisted on so many revisions leading a (very) long reviewing process. I hope that going through this process will help the authors reach a sufficient level of rigour and detail in future manuscripts, before their submission.